# Reward Learning from Multiple Feedback Types

**Yannick Metz[1,2], András Geiszl[2], Raphaël Baur[2], Mennatallah El-Assady[2]**
[1]University of Konstanz, Germany
`yannick.metz@uni-konstanz.de`
[2]ETH Zurich, Switzerland
`ageiszl@inf.ethz.ch, {raphael.baur, menna.elassady}@ai.ethz.ch`

## Abstract

Learning rewards from preference feedback has become an important tool in the alignment of agentic models. Preference-based feedback, often implemented as a binary comparison between multiple completions, is an established method to acquire large-scale human feedback. However, human feedback in other contexts is often much more diverse. Such diverse feedback can better support the goals of a human annotator, and the simultaneous use of multiple sources might be mutually informative for the learning process or carry type-dependent biases for the reward learning process. Despite these potential benefits, learning from different feedback types has yet to be explored extensively. In this paper, we bridge this gap by enabling experimentation and evaluating multi-type feedback in a broad set of environments. We present a process to generate high-quality simulated feedback of six different types. Then, we implement reward models and downstream RL training for all six feedback types. Based on the simulated feedback, we investigate the use of types of feedback across ten RL environments and compare them to pure preference-based baselines. We show empirically that diverse types of feedback can be utilized and lead to strong reward modeling performance. This work is the first strong indicator of the potential of multi-type feedback for RLHF.

## 1 Introduction

Reinforcement learning from human feedback (RLHF) is a powerful tool to train agents when it is difficult to specify a reward function or when human knowledge can improve training efficiency. Recently, using multiple forms of human feedback for reward modeling has come into focus (Jeon et al., 2020; Ghosal et al., 2023; Ibarz et al., 2018; Bıyık et al., 2022a; Mehta & Losey, 2022). Using diverse sources of information opens up several possibilities: (1) feedback from different sources allows for correcting potential biases in the data; (2) the feedback type can be adapted to a particular task or user based on preferences, knowledge state, or available input modalities; (3) agents can actively select an appropriate type of feedback during training to optimize learning (Jeon et al., 2020). However, there are only a few empirical investigations on the characteristics of different feedback types (Mehta & Losey, 2022; Ghosal et al., 2023; Bıyık et al., 2022a). Even though there have been some efforts to unify different types of reward (Jeon et al., 2020), and attempts to learn from a combination of two or three feedback types (Ibarz et al., 2018; Mehta & Losey, 2022), the use of a larger number of different feedback types has not been fully explored. This paper addresses three key questions: (1) How can we define, model, and simulate different explicit types of human feedback consistently? (2) What characteristics do reward functions learned from these feedback types exhibit? (3) Can we combine reward models from different feedback types for robust learning?

**Contributions** — To answer these, we contribute by (1) Implementing synthetic generation and reward models for six distinct human feedback types, along with a joint-reward modeling approach based on a reward function ensemble (section 3), (2) Empirically investigating the effectiveness and complementarity of these feedback types (section 4), and (3) Analyzing the performance of joint training with multiple feedback types, highlighting the potential of learning from diverse human feedback in future applications (section 5).

## 2    RELATED WORK

**Reinforcement Learning from Human Feedback —** Using human feedback as the sole or an additional source of reward information has gained traction in research (Ng et al., 2000; Knox & Stone, 2009; Griffith et al., 2013; Christiano et al., 2017), especially for the alignment of large language models (Ouyang et al., 2022). This paper focuses on learning rewards solely from human feedback, excluding approaches like interactive reward shaping (Knox & Stone, 2009; Warnell et al., 2017).

**Synthetic Generation of Human Feedback —** So far, the generation of synthetic feedback has been primarily explored in the area of preferences (Bıyık et al., 2022b; Christiano et al., 2017). In their foundational work for modern RLHF, (Christiano et al., 2017) utilize a simple oracle model based on underlying ground truth reward, which uses the sum of rewards over segments to compute preferences. To account for human error, they assume a fixed miss-labeling probability of $10\%$, implemented via preference label switching. (Bıyık et al., 2022b) implement a comprehensive toolbox to simulate preferences with adaptable levels of noise/irrationality and methods to collect data from humans. We extend these efforts into a more versatile multi-type feedback setting.

**Combining Multiple Reward Functions and Multi-Type Feedback —** Different types of feedback, ranging from ratings, demonstrations (Ng et al., 2000; Abbeel & Ng, 2004), comparisons (Wirth et al., 2017) to interruptions (Hadfield-Menell et al., 2017) or even language and narrations (Fish et al., 2018; Sumers et al., 2022b) has been proposed. A wide range of different types of feedback can be interpreted via a common framework of reward-rational choice (Jeon et al., 2020). There has been some work on combining multiple feedback types, most notably demonstrations and preferences Ibarz et al. (2018); Bıyık et al. (2022a). However, the authors used separate phases of demonstrations and preference-based learning. Most similar in spirit is work by Mehta & Losey, who also train a single reward model based on the reward rational choice framework. However, they restrict themselves to two (three) feedback types, including demonstrations and preferences (as well as stops).
Sumers et al. investigate evaluative, instructive, and descriptive language feedback in a simple bandit environment. We see our work as an extension of these efforts towards more complex environments and more diverse and dynamic feedback.
Recently, there has been some work on extending the space of possible feedback by presenting implementations for collection from human annotators (Metz et al., 2023; Yuan et al., 2024). However, these works do not investigate the full implementation of diverse reward models and have not analyzed reward learning and RL from diverse feedback.

## 3    DEFINING AND SYNTHESIZING MULTI-TYPE FEEDBACK

We start our investigation by defining relevant feedback types, describing our approach to simulate feedback of these different types, and discussing the implementation of reward models.

**Preliminaries —** We assume a RLHF scenario with an agent that acts within an environment, observing states $s_t \in \mathcal{S} \subseteq \mathbb{R}^N$ and performing actions $a_t \in \mathcal{A} \subseteq \mathbb{R}^M$. The agents select actions according to a policy $\pi$, which throughout training is optimized with respect to a learned reward function $\hat{r} : \mathcal{S} \times \mathcal{A} \to \mathbb{R}$. In RLHF, the reward function estimator $\hat{r}$ is updated based on human feedback. In RLHF, humans often provide feedback over trajectories $\xi \in \Xi$, with $\Xi$ being the set of all possible trajectories. Trajectories are sequences of states and actions, e.g., generated by the agent acting within the environment. Providing feedback for trajectories requires fewer human interactions, as human labelers do not need to label every state-action pair. Finally, in this paper, we refer to an expert policy $\pi_e$. $\pi_e$ is the policy that maximizes the true reward function, which is not observable in humans. We use $\pi(\xi_{t:t+H}) \coloneqq \prod_t^{t+H} \pi(s_t)$ as a shorthand for the probability of a policy acting over a segment.

### 3.1    FEEDBACK TYPES

In the scope of this study, we decided on six exemplary types of feedback motivated by existing work (Metz et al., 2023). We briefly present how these six types of feedback can be defined individually.

### 3.1.1 EVALUATIVE FEEDBACK

**F1: Rating Feedback** — We define this type of feedback for cases in which a human gives a numerical or otherwise quantifiable judgment of a target, e.g., binary feedback or a numeric score (Arzate Cruz & Igarashi, 2020). We define *evaluative feedback* as a mapping from a target $\mathcal{T}$ to a scalar value in a value set $v_{fb} \in V \subseteq \mathbb{R}$. Common choices for the value set can be a binary score $V = -1, 1$ or a rating $V = 1, .., 10$, which we choose for our experiments:

$$\mathcal{F}_{eval}(\xi) = v_{fb} \in V = \{1, 2, 3, ..., 10\}$$

**F2: Comparative Feedback** — Here, a user makes a relative judgment, i.e., a pairwise comparison between two targets or a ranking of multiple targets. This type of preference-based feedback is widely used because it is assumed that it is easier for humans to give comparative judgment compared to absolute scores (Wirth et al., 2017). Comparative feedback is a set of targets with an associated ordering relation. We define an instance of comparative feedback as a preference over two segments:

$$\mathcal{F}_{comp}(\xi_1, \xi_2) = v_{fb} \in \{\succ, \prec, =\}$$

### 3.1.2 INSTRUCTIONAL FEEDBACK

**F3: Demonstrative Feedback** — The human is asked to provide a reference of optimal behavior that the agent should imitate (Ng et al., 2000). We generally assume that these demonstrations are optimal w.r.t. the "internal" human reward function. A demonstration is not conditioned on any specific trajectory but rather independent and is given for the entire trajectory space.

$$\mathcal{F}_{demo}(\Xi) := \bar{\xi} \quad \text{with} \quad \bar{\xi} \sim \pi_e$$

with $T$ being a final state (e.g., a terminal environment state).

**F4: Corrective Feedback** — Here, the user has a trajectory showcasing imperfect behavior as a reference and needs to provide an improved trajectory. This can be done either implicitly, e.g., by pushing a robot into a correct position (Mehta & Losey, 2022), or explicitly via specifying a better action. We define a correction as:

$$\mathcal{F}_{corr}(\xi_{t:t+H}) = \Delta\xi_{t:t+H}, \quad \text{with} \quad \xi_{t:t+H} + \Delta\xi_{t:t+H} \succ \xi_{t:t+H}$$

### 3.1.3 DESCRIPTIVE FEEDBACK

**F5: Descriptive Feedback** — The definition of descriptive feedback is more open, and we could interpret any declarative statement about an MDP as descriptive (Rodriguez-Sanchez et al., 2023). We chose a formulation in which descriptive feedback is a qualitative judgment about features, compared to (Sumers et al., 2022a), i.e., information about "state-action pairs like these are..." As such, descriptive feedback should not inform about a single trajectory but instead, serve as a way to assign reward information about a set of features:

$$\mathcal{F}_{desc}(f_s \subseteq \mathcal{S}, f_a \subseteq \mathcal{A}) = v_{fb} \in \mathbb{R}$$

**F6: Descriptive Preferences** — As an extension to descriptions that mirror comparative and corrective feedback, which can be interpreted as relative versions of evaluative and demonstrative feedback, respectively, we propose descriptive preferences. Descriptive preferences are similar to feature preferences proposed before Yuan et al. (2024).

$$\mathcal{F}_{des.pref.}((f_s^1 \subseteq \mathcal{S}, f_a^1 \subseteq \mathcal{A}), (f_s^2 \subseteq \mathcal{S}, f_a^2 \subseteq \mathcal{A})) = v_{fb} \in \{\succ, \prec, =\}$$

These six feedback types are a non-exhaustive selection of human feedback, and many more can be imagined (Jeon et al., 2020; Kaufmann et al., 2024; Metz et al., 2023). However, we argue that our selection represents a set of fundamental, general, and explicit human feedback types. Many other types of human feedback are implicit (Kovač et al., 2021; Kaufmann et al., 2024), i.e., need to be reduced to an explicit reward signal eventually, or collect meta-information that support the interpretation of human feedback (Kaufmann et al., 2024).

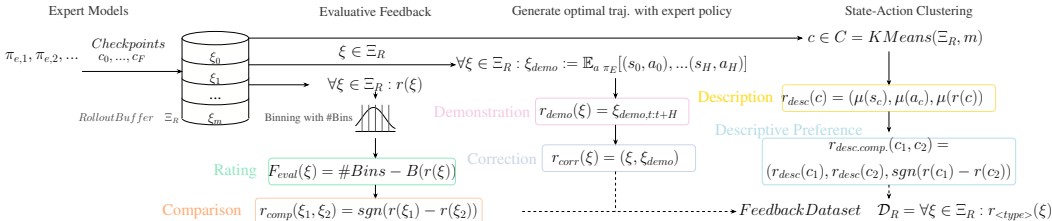

Figure 1: Generation of Simulated Feedback of different types: Based on an existing expert model and rollout buffer, we generate six types of feedback, including ratings and comparisons, demos and corrections, as well as descriptions and descriptive preferences.

## 3.2 SIMULATING MULTI-TYPE HUMAN FEEDBACK

We base our experimental evaluation on simulated feedback of different types to enable larger-scale environments and ensure reproducibility. While there exists some work to simulate individual feedback types (e.g., preference feedback for locomotion environments, Atari (Brown et al., 2019b) or language modeling (Dubois et al., 2024)), pipelines to provide feedback for multiple feedback types has been limited so far (Yuan et al., 2024; Metz et al., 2023). Yuan et al. describe five feedback types but only collected annotated datasets for two feedback types.

To tackle these challenges, we implement a lightweight [*] software library to generate feedback of six types mentioned above. The library is fully interoperable with well-established RL frameworks such as *Gymnasium* (Towers et al., 2024), *Stable Baselines* (Raffin et al., 2021), and *Imitation* (Gleave et al., 2022). It provides utilities for synthetic feedback generation, reward models, and agent training.

### 3.2.1 GENERATION PROCESS

We describe a method to generate fixed labeled feedback datasets for reward model training. Our approach can also be adapted for dynamic feedback generation with minimal modifications. Figure 1 gives an overview of the synthetic generation process. We aim to create synthetic feedback for trajectory data based on the six abovementioned types.

As the foundation for feedback datasets, we sample segments across 20 model checkpoints from four expert RL models for each environment to ensure diversity, with random start indices and truncation at the end of episodes (Details in subsection B.6).

**Evaluative Feedback** — We simulate ratings (1-10) based on discounted segment returns, creating a more realistic scenario than raw rewards. Using a calibration set $\Xi_{cal} := \xi_0, \xi_1, ...,$ we implement equal-width binning of discounted returns, assigning scores decrementally from highest (10) to lowest (1) bins (distribution shown in Figure 11). Online feedback generation compares incoming returns against this binning, with optional dynamic updates to maintain distribution consistency. For *comparative feedback*, we derive pairwise comparisons from reward differences, excluding pairs with similar rewards (difference $< 10\%$ of return standard deviation).

**Instructive Feedback** — For both corrective and demonstrative feedback, we utilize best-performing expert checkpoints. We generate expert trajectories from saved initial states using policy $\pi_e$, selecting the highest-performing samples from multiple expert models. For **corrective feedback**, we retain only corrections achieving higher discounted returns than original segments (comparison in Figure 15). When state resets are unavailable, single-step corrections via expert action queries provide an alternative. **Demonstrative feedback** utilizes these high-quality corrections independently.

**Descriptive Feedback** — We implement clustering using mini-batch k-means on concatenated state-action pairs. The number of clusters matches other feedback types' sample sizes, providing comparable trajectory space resolution. Each cluster's mean representative serves as the reward model target, with averaged rewards as description scores. **Descriptive preferences** are derived from reward differences between randomly sampled cluster pairs.

---

[*]The code is available at: `https://github.com/ymetz/multi-type-feedback`

### 3.3 Introducing Feedback Noise

We implement a consistent noise scheme across feedback types based on human feedback rationality modeling (Jeon et al., 2020; Ghosal et al., 2023). To ensure consistency, rather than directly introducing noise for each individual feedback type, we introduce noise into the underlying reward distribution, which then influences generated feedback.

For evaluative feedback, we add noise via a truncated Gaussian distribution $\mathcal{N}_T(\mu, \sigma, lower, upper)$:
$$f_{fb,eval} = f_{fb} + \mathcal{N}_T(\mu = 0, \sigma = \beta * 10, 1, 10)$$
Preference labels are flipped according to perturbed rewards, effectively reducing rationality by increasing incorrect preference probability. More nuanced than random flips, this approach primarily affects segments with similar rewards.

For descriptive feedback, the noise scale is determined by cluster reward ranges:
$$f_{fb,descr} = r(c) + \mathcal{N}_T(\mu = 0, \sigma = \beta * |max(r_c) - min(r_c)|, min(r_c), max(r_c))$$

For demonstrative feedback, we perturb states and actions based on the observed standard deviations in the dataset:
$$\bar{\xi} = (s_i + \mathcal{N}_T(\mu = 0, \sigma = \beta * \sigma(s)), a_i + \mathcal{N}_T(\mu = 0, \sigma = \beta * \sigma(a)))_{t:t+H}$$

In Appendix B, we detail the feedback dataset generation process and its characteristics to ensure reproducibility. We see the documentation of the generated datasets as a key part of our approach, as it influences down-stream analysis. An alternative regret-based scheme is discussed in subsection B.5. Real human feedback may contain various biases, and we encourage human-subject studies to determine appropriate values for different feedback types.

### 3.4 Reward Functions from Feedback Types

For our baseline implementation of single reward functions, we closely follow the established methodology from RLHF: For feedback mapping to scalar values (*Ratings* and *Descriptions*), our reward model is optimized with an MSE-loss between the predicted reward $\hat{r}$ and annotated scalar feedback values $v_{fb}$. For relative feedback (such as *Comparisons*, *Corrections* and *Descriptive Preferences*), we optimize the reward function with a cross-entropy loss between predicted and annotated pairwise preferences.

$$\mathcal{L}_{MSE}(\hat{r}) = \frac{1}{n} \sum_{\xi \in \mathcal{D}} (v_{fb}(\xi) - \sum_{s \in \xi} \hat{r}(s))^2 \tag{1}$$

$$\mathcal{L}_{MLE}(\hat{r}) = - \sum_{(\xi^1, \xi^2, \mu) \in \mathcal{D}} \mu(1) \log \hat{P}[\xi^1 \succ \xi^2] + \mu(2) \log \hat{P}[\xi^2 \succ \xi^1]. \tag{2}$$

with $\hat{P}[\xi^1 \succ \xi^2]$ following the established Bradley-Terry model (Bradley & Terry, 1952):
$$\hat{P}[\xi^1 \succ \xi^2] = \frac{\exp \beta \sum_i \hat{r}(o_i^1, a_i^1)}{\exp \beta \sum_i \hat{r}(o_i^1, a_i^1) + \exp \beta \sum_i \hat{r}(o_i^2, a_i^2)} \tag{3}$$

To train the reward model, we also model *Demonstrative* feedback as preferences between the optimal demonstrations and trajectories based on random policies, i.e., random behavior. In our experiments, sampling against random behavior has been stronger than sampling against other sub-optimal rollouts. We follow the reward (rational) choice formalism (Jeon et al., 2020) for relative feedback. We discuss the formalization of our proposed feedback types under this framework in Appendix A. For our baseline experiments, we assume optimal reward, i.e., a rationality coefficient of $\beta = \infty$. However, we will investigate learning from sub-optimal, more irrational feedback later. In line with existing work (Christiano et al., 2017), we optimize our reward model over trajectory segments $\xi$, with a maximal length of 50 steps, except for descriptive feedback, which uses single states.

In our analysis, we pre-train reward models based on trajectories collected from different checkpoints of an RL model, similar to (Brown et al., 2019a). This approach should lead to the model learning the reward distribution for a representative set of state-action pairs. For details, we refer to Appendix D.

| | | |
|---|---|---|
| $(\xi, v_{fb}), \mathcal{L}_{MSE}$ | $(\xi_i, \xi_j, \prec), \mathcal{L}_{MLE}$ | $(\xi_{rand}, \bar{\xi}_i, \prec), \mathcal{L}_{MLE}$ |
| *Ratings* | *Comparisons* | *Demonstrations* |
| $(\xi_i, \xi_i^e, \prec), \mathcal{L}_{MLE}$ | $(s_c, a_c, r_c), \mathcal{L}_{MSE}$ | $((s_{c,1}, a_{c,1}), (s_{c,2}, a_{c,2})), \prec), \mathcal{L}_{MLE}$ |
| *Corrections* | *Descriptions* | *Descriptive Comparison* |

Table 1: A summary of the reward model implementations for different feedback types, i.e., the reward inputs and loss function used to optimize the reward model.

# 4 INVESTIGATING THE EFFECTIVENESS OF DIFFERENT FEEDBACK TYPES

We investigate the effectiveness and characteristics of reward learning across different feedback types, focusing on three key research questions: (1) How do various feedback types compare to sampled pairwise comparisons under both optimal and perturbed conditions? (2) What distinct characteristics emerge in reward functions and agent behavior across different feedback types? (3) Does combining multiple feedback types offer advantages over single-type approaches? We explore this through an ensemble of feedback-type reward functions in RL agent training.

## 4.1 TRAINING SETUP

We evaluate our approach on established benchmark environments: *Mujoco* locomotion environments (*HalfCheetah-v5, Walker2d-v5, Swimmer-v5, Ant-v5, Hopper-v5, Humanoid-v5*), a *MetaWorld* Yu et al. (2019) environments (*sweep-into-v2*), and three discrete action space environment from *HighwayEnv* Leurent (2018): *merge-v0, highway-fast-v0, roundabout-v0*.

For expert model training, we use PPO (Schulman et al., 2017) and SAC (Haarnoja et al., 2018), implementing tuned hyperparameters from available implementations (Raffin, 2020; Cobbe et al., 2019; Machado et al., 2018). The reward model consists of a six-layer MLP with ReLU activations.

Our feedback datasets comprise 10000 segments of 50 steps or 10000 descriptive clusters. We generate five separate feedback datasets to control for dataset composition variations due to random sampling and environment stochasticity. Reward models undergo supervised pre-training on labeled datasets (see Appendix D).

We evaluate learned reward functions using a simplified RLHF setup with a single pre-trained reward model. Agent training follows a standard PPO configuration (details in Appendix E), with results aggregated across five random seeds unless specified otherwise. As an additional baseline, we train policies via behavioral cloning on the collected demonstration datasets (details in subsection E.5).

## 4.2 RL WITH LEARNED REWARD FUNCTIONS FROM OPTIMAL FEEDBACK

Initial reward model training across all feedback types shows effective learning with decreasing validation loss curves (detailed in Appendix D). All reward types demonstrate steady loss reduction even when noise is introduced, with specific effects discussed in the following section.

Figure 2 demonstrates that different feedback types achieve competitive performance across environments. Some feedback types can match the expert model performance for three environments with ground-truth feedback. However, performance is significantly worse in other environments like *Swimmer-v5* or *Ant-v5* than training with a ground-truth reward function. Descriptive feedback and descriptive preferences generally yield the best results, though specific environments show exceptions—for instance, demonstrative feedback outperforms others in *Swimmer-v5*.

## 4.3 SENSITIVITY TO FEEDBACK VALUE NOISE

While our initial analysis assumed optimal feedback relative to ground-truth rewards, real human feedback inevitably contains errors. We investigate this by introducing controlled noise into the feedback data, using comparable perturbation scales across feedback types.

Figure 3 illustrates varying sensitivity to noise across feedback types. While the validation loss of reward function decreases across feedback types, downstream RL performance varies by feedback

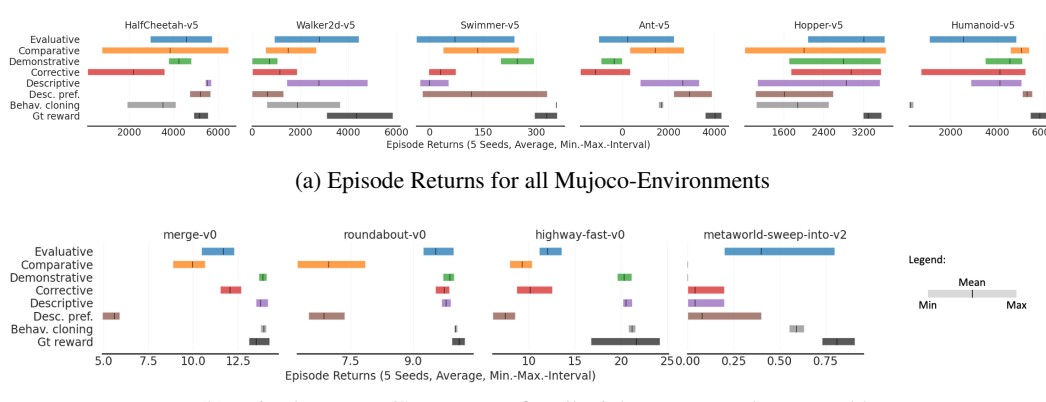

(a) Episode Returns for all Mujoco-Environments

(b) Episode Returns/Success rate for all Highway-Env and Metaworld

Figure 2: RL from individual feedback type reward models. The area boundaries indicate minimum and maximum values out of the sampled runs. Results are averaged over five random seeds/ feedback datasets. For full training curves see subsection E.2.

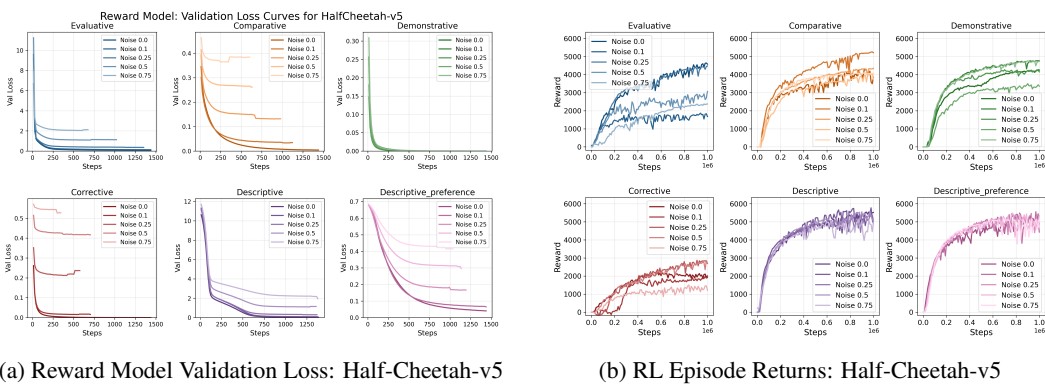

(a) Reward Model Validation Loss: Half-Cheetah-v5

(b) RL Episode Returns: Half-Cheetah-v5

Figure 3: Showcasing the influence of noise on different feedback types in the **HalfCheetah-v5** environment.

type. For *Half-Cheetah-v5* feedback seems more susceptible, whereas descriptive feedback is more robust. Notably, moderate noise can sometimes improve downstream RL performance despite negatively affecting reward learning, particularly for corrective feedback. We hypothesize this acts as a form of label smoothing, encouraging the model to focus on more distinctly different pairs rather than highly similar ones affected by noise. We provide additional results in subsection E.3.

**Summary: —** Our experiments demonstrate that various feedback types are effective across different environments and training stages. Comparative feedback is matched or even surpassed by other types across environments. While descriptive feedback shows consistently strong performance, its real-world collection presents challenges, though approaches using state clustering have been proposed Zhang et al. (2022). These findings suggest potential benefits in leveraging multiple feedback types to exploit their respective strengths in specific scenarios.

## 4.4 ANALYZING REWARD FUNCTIONS FROM DIFFERENT FEEDBACK TYPES

Beyond downstream RL agent performance, we analyze the learned reward functions to better understand the characteristics of different feedback types.

**Correlation with Ground-Truth Reward Function —** Figure 4 illustrates pairwise correlations between ground-truth and learned reward functions, computed on a separate validation dataset aggregated across five feedback datasets.

Most feedback types demonstrate a high correlation with the ground-truth reward function in Mujoco,

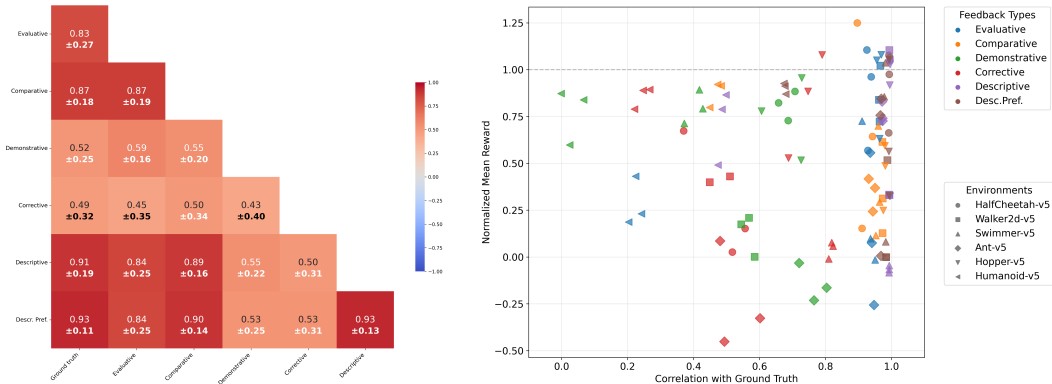

(a) Aggregated reward function correlations (Mujoco, five seeds, optimal). Details see subsection D.5

(b) Relationship between the correlation of ground-truth reward function, learning reward function, and downstream reward (normalized by expert rewards).

Figure 4: Correlation of Learned Reward Functions for Different Feedback Types: (a) Correlation differs between types (b) We do not observe a strong relationship between GT correlation and rewards.

except for corrections and demonstrations, which show greater divergence due to their dependence on expert policy rather than direct derivation from ground-truth. As shown in Figure 4(b), the weak correlation between ground-truth reward similarity and final reward suggests that alternative feedback types can uncover different yet effective reward functions. Furthermore, high correlation is itself not indicative of downstream performance.

**Effectiveness for Different Noise Levels —** Figure 5 demonstrates varying noise sensitivity across feedback types in two environments. Increased noise generally reduces correlation with ground-truth rewards. Descriptive feedback shows more stability to noise, consistent with its robust downstream RL performance. In contrast, evaluative and comparative rewards exhibit larger performance drops. Demonstrative feedback has a lower correlation and is less affected by noise. These effects are also visible across other environments subsection D.6.

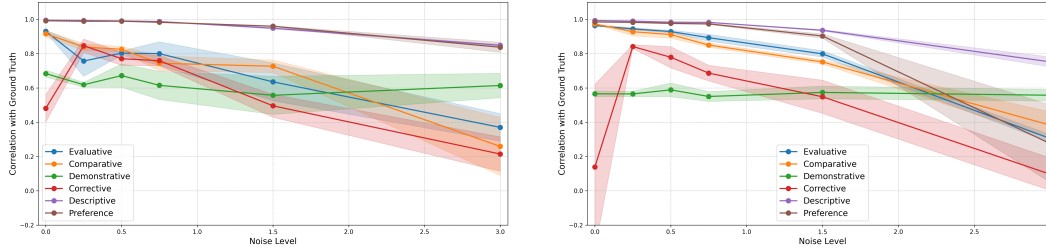

(a) Correlation with ground truth reward function over different noise levels for **HalfCheetah-v5**

(b) Correlation with ground truth reward function over different noise levels for **Walker-v5**

Figure 5: Robustness of different reward functions for different noise levels: With increasing noise levels, we see a drop in performance for types, but to a different degree.

**Comparing Predictions over Sequences —** Analyzing reward predictions across 100-step trajectories reveals interesting patterns. Figure 6 demonstrates that a high correlation with ground-truth rewards does not necessarily translate to strong RL performance. For instance, models achieve near-expert performance in the Humanoid environment despite low reward correlation, particularly with demonstrative feedback. This suggests that effective RL training can occur with various reward functions independent of their ground-truth correlation.

**Summary: —** Our analysis reveals that most feedback types achieve high ground-truth accuracy, excluding demonstrative and corrective. However, even uncorrelated reward functions, like demonstrative feedback, can be highly informative and enable learning in downstream RL. The varying performance characteristics across feedback types, especially with introduced noise, suggest comple-

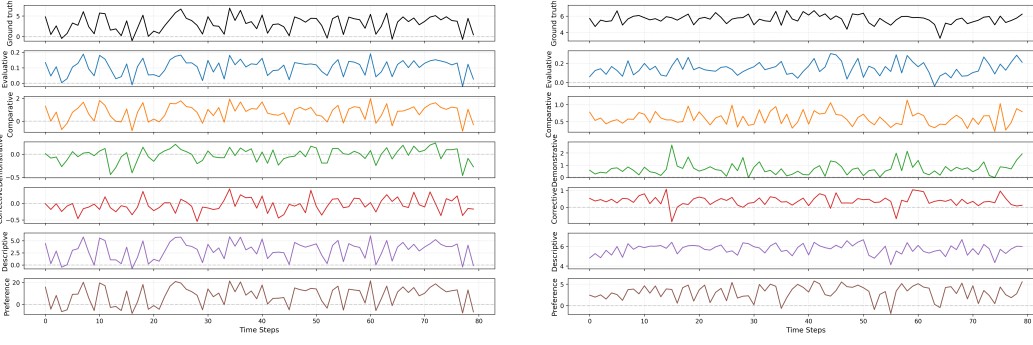

(a) Sequential reward predictions for **Ant-v5**     (b) Sequential reward predictions for **Humanoid-v5**

Figure 6: Reward predictions from learned reward functions: (a) *Ant-v5* shows high correlation with ground-truth rewards yet suboptimal agent performance. (b) *Humanoid-v5* maintains strong downstream performance despite lower reward correlations than other Mujoco environments.

mentary strengths that could be combined.

## 5    JOINT-MODELING TO LEARN REWARDS FROM MULTI-TYPE HUMAN FEEDBACK

While different feedback types effectively learn reward functions, their performance varies across scenarios and noise conditions. We explore a proof-of-concept approach combining multiple feedback types through ensemble methods. **Averaging Model —** Using our pre-trained reward models, we implement a basic ensemble that averages predictions from all six feedback types. **Uncertainty-Weighted Ensemble —** We extend the basic ensemble by incorporating epistemic uncertainty to mitigate underfitting. Using standard deviations from individual reward model ensembles Durasov et al. (2021), we weigh each feedback type's predictions based on model inverse uncertainty to leverage complementary strengths.

### 5.1    RESULTS

Figure 7 demonstrates environment-specific success. The ensemble matches the best individual feedback performance in *HalfCheetah-v5*, significantly outperforming the single-feedback average. However, *Walker2d-v5* shows limited improvement, performing at or below average, indicating room for improvement in complementary learning. These results indicate that the combination of feedback types might harness the information within the set of feedback types. However, it might fail in other scenarios.

Our preliminary experiments (see subsection C.2) suggest that selective feedback type combinations outperform full ensembles, opening promising directions for optimizing feedback type selection and combination strategies. As shown in Figure 7, this approach has limitations, including potentially improper uncertainty calibration, though we expect it to provide full utility and present additional challenges in an online learning setup where specific feedback types might be queried, and we strongly encourage further exploration in this research area.

## 6    DISCUSSION AND OUTLOOK

Our work yields two key insights: (1) Multiple feedback types effectively enable reward learning and RL, with noise-robust results across environments, challenging the presumed superiority of comparative feedback. (2) While simultaneous learning from multiple reward models shows promise, further research is needed to realize its potential fully.

Our implementation advances multi-type feedback beyond theoretical frameworks (Jeon et al., 2020). We identify several key directions for future work:

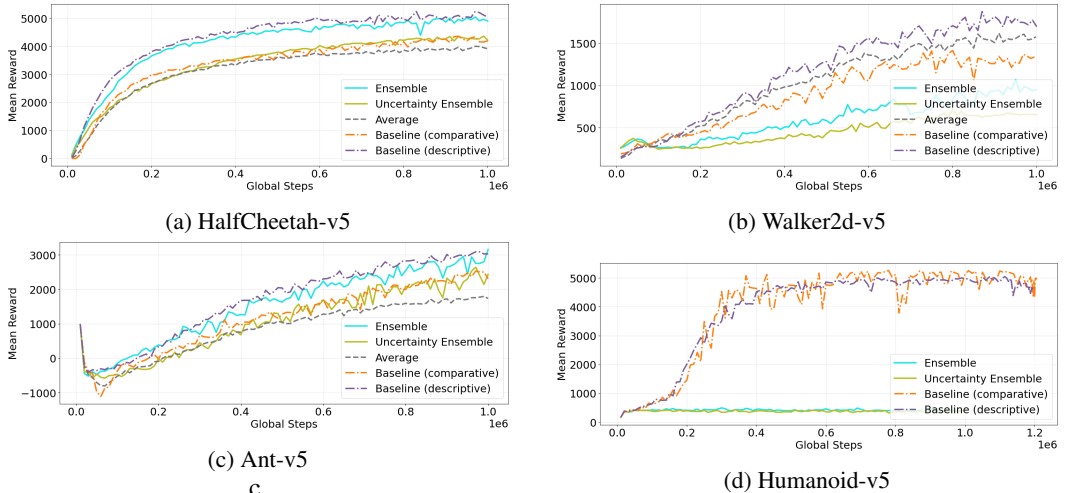

Figure 7: Comparison between the ensemble (with and without uncertainty-based scaling), average returns across all single feedback RL runs, descriptive and comparative baselines (averaged over 12 runs). The joint modeling approach can match the best single type but also has failure cases.

**Dynamic Reward Models —** Transitioning from fixed pre-trained to continuously updating models could enable more dynamic learning processes. This approach could incorporate complex querying mechanics Zhan et al. (2023); Jeon et al. (2020); Metz et al. (2023) and leverage different feedback types at various training stages, building on our weighted ensemble implementation.

**Scaling to Complex Domains —** While our evaluation spans multiple RL environments, extending these findings to more complex domains remains crucial, particularly LLM alignment and fine-tuning. Additionally, expanding to more feedback types presents opportunities and challenges—while integration is straightforward, efficient strategies for managing multiple feedback sources need investigation.

**Human Feedback Integration —** A natural extension is transitioning to real human feedback, where adapting synthetic reward generation to observed human feedback patterns could improve experimental validity. In particular, we base our simulation on simple assumptions, such as modeling noise via an additive Gaussian distribution. Fitting characteristics to measurements from human experiments promises more realistic experiments in the future.

**Exploring RL from AI Feedback —** Strongly related to the concept of synthetic feedback discussed in this paper is the use of RL from AI Feedback (RLAIF) Lee et al. (2024); Bai et al. (2022). As human feedback annotations may be hard to acquire at scale, RLAIF based on limited human guidance can vastly improve efficiency Liu et al. (2024); Lee et al. (2024). Multi-type feedback can enable new ways of integrating various sources of human and AI feedback. Our framework allows the investigation and comparison of new sources and comparison of their effectiveness.

## 7 CONCLUSION

This paper demonstrates the effectiveness of multi-type human feedback for reward learning, moving beyond traditional binary preferences. We defined six feedback types and described a protocol to generate synthetic feedback. We systematically evaluate learned reward functions and reveal some of their characteristics and complementary strengths in reward learning and downstream RL performance. Our analysis of joint reward models suggests that joint modeling can perform well but needs further exploration. We encourage researchers to explore more diverse human-AI interaction paradigms and contribute to the development of more capable and reliable AI systems.

REPRODUCIBILITY STATEMENT

Source code for all experiments is available in the supplementary materials, and will be open sourced. We used the *Stable-Baselines3* Raffin et al. (2021) library (Version 2.3.2) to train all RL expert models with the provided default architectures for MLP- and CNN-based policies. Furthermore, we used the *RL Baselines3 Zoo* Raffin (2020) framework (Version 2.3.2) to train the agents, and used the included hyperparamters to train both the expert models and downstream RL models with learned reward models. Hyperparameters and scores of expert models are reported in subsection B.6. Reward models were implemented and trained as *Pytorch Lightning*-models, with the training and architecture details reported in Appendix D.

All experiments were performed on machines with single *Nvidia V100* GPUs. Most tasks were CPU-bound and required relatively little GPU memory. We estimate that reproducing all experiments results reported in the paper, takes approximately 500 hours on a single node with 4 GPUs and 64 cores. The runtime for feedback generation of 10,000 steps is approximately around 60 minutes (depending on network inference time and environment complexity), fitting of reward models takes 10-30 minutes. The training times of (expert) RL models are highly dependent on the scenario (timesteps and availability of parallel simulation environments). With a learned reward function, training of a Mujoco environment with SAC and 1 million environment steps takes around one hour. Training with a reward model ensemble does only impose a modest runtime increase, but potentially a large memory increase (if all reward models are kept in GPU memory).

ACKNOWLEDGEMENTS

The authors acknowledge support by the state of Baden-Württemberg through bwHPC. Part of this research was conducted with financial support from a DAAD Research grants for doctoral students.

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

## A  DETAILS ON THE FORMALIZATION OF FEEDBACK TYPES

This study investigates six exemplary feedback types motivated by existing work (Metz et al., 2023). Here, we will briefly model these feedback types according to the reward rational choice formalism presented in previous work (Jeon et al., 2020). Table 2 and Table 3 show the choice set, grounding functions and constraint-based formulations respectively. The formalism proposes that feedback types can be described by an explicit or implicit set of (feedback) choices, that $\mathcal{C}$ *humans* optimize over, and a grounding function $\psi : \mathcal{C} \to f_{\Xi}$, which maps these choices $c \in \mathcal{C}$ to a distribution over trajectories.

Table 2: Feedback types, choice sets, and grounding functions

| Feedback type | $\mathcal{C}$ (Choice set) | $\psi : \mathcal{C} \to \Xi$ (Grounding func.) |
|---|---|---|
| **Rating** | $(\xi, f_{fb}) \in \{\xi\} \times [a,b], [a,b] \subset \mathbb{R}$ | $\psi(\xi, f_{fb}) = \xi$ |
| **Comparative** | $\xi_i \in \{\xi_1, \xi_2\}$ | id |
| **Corrective** | $(\xi, \Delta\xi) \in \{\xi\} \times \{\xi_c - \xi \mid \xi_c \in \Xi\}$ | $\psi(\xi, \Delta\xi) = \xi + \Delta\xi$ |
| **Demonstrative** | $\xi_d \in \Xi$ | id |
| **Descriptive** | $(f_s \in \mathcal{S}, f_a \in \mathcal{A}, r \in \mathbb{R})$ | $\{(s,a) \in \Xi \mid s = f_s, a = f_a\}$ |
| **Descr. Pref.** | $(f_{s,i}, f_{a,i}) \in \{(f_{s,1} \in \mathcal{S}, f_{a,1} \in \mathcal{A}), (f_{s,2} \in \mathcal{S}, f_{a,2} \in \mathcal{A})\})$ | $\{(s,a) \in \Xi \mid s = f_s, a = f_a\}$ |

Table 3: Feedback types and corresponding constraints

| Feedback type | Constraint |
|---|---|
| **Evaluative** | $r(\xi) = f_{fb}$ |
| **Comparative** | $\begin{cases} r(\xi_1) \geq r(\xi_2) & \text{if} \quad \xi_i = \xi_1 \\ r(\xi_2) \geq r(\xi_1) & \text{if} \quad \xi_i = \xi_2 \end{cases}$ |
| **Corrective** | $r(\xi + \Delta\xi) \geq r(\xi)$ |
| **Demonstrative** | $r(\xi_d) \geq r(\xi) \quad \forall \xi \in \Xi$ |
| **Description** | $r(\{(s,a) \in \Xi \mid s = f_s, a = f_a\}) = v_{fb}$ |
| **Descriptive Preference** | $\begin{cases} r(\{(s,a) \in \Xi \mid s = f_{s,1}, a = f_{a,1}\}) \geq r(\{(s,a) \in \Xi \mid s = f_{s,2}, a = f_{a,2}\}) & \text{if} \quad i = 1 \\ r(\{(s,a) \in \Xi \mid s = f_{s,2}, a = f_{a,2}\}) \geq r(\{(s,a) \in \Xi \mid s =_{s,1}, a = f_{a,1}\}) & \text{if} \quad i = 2 \end{cases}$ |

## B  ANALYSIS OF SYNTHETIC GENERATED FEEDBACK

In this first appendix, we want to highlight the characteristics of different types of synthetic feedback. We want to discuss details because the simulated feedback is integral to our investigation.

### B.1  DETAILS OF EVALUATIVE FEEDBACK

Evaluative feedback is based on discounted segments returns (i.e., the discounted sum of step rewards within a segment). The discount factors $\gamma$ match the discount factors of the expert models (as reported in subsection B.6). Figure 8 and Figure 9 show the distribution of discounted segments returns for the feedback datasets. The distribution of rewards is a result of the data collection strategy, i.e. collecting segments from a set of checkpoints throughout training. To generate ratings, we use an equal-width binning approach of returns, and assign ratings of 1 to 10 for each bin.

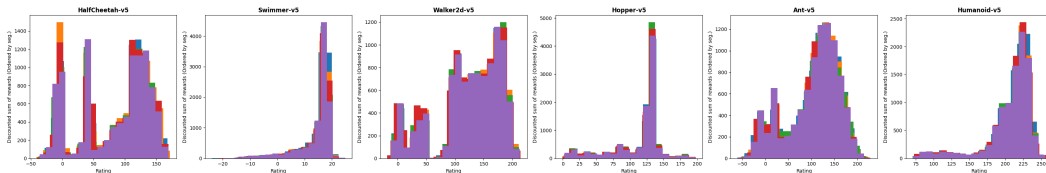

Figure 8: Histogram of total discounted reward distribution (for 10000 randomly sampled segments with max. length 50/ truncated at episode end). Each color represents a different generated feedback dataset.

---

**Algorithm 1** Generation of Evaluative and Comparative Feedback

---

**Require:** Environment $E$, Expert models $\mathcal{E} = \{e_1, \ldots, e_n\}$, Novice model checkpoints $\Theta = \{\theta_1, \ldots, \theta_m\}$, Steps per checkpoint $T$, Segment length $L$, Discount factor $\gamma$

1:   $\mathcal{S} \leftarrow \varnothing$          ▷ Set of segments
2: **for** $\theta \in \Theta$ **do**
3:     $\pi_\theta \leftarrow \text{NoviceModel}(\theta)$
4:     $s_0 \leftarrow E.\text{reset}()$
5:     $F \leftarrow \varnothing, C \leftarrow \varnothing$       ▷ Feedback and state copy collection
6:     **for** $t = 1$ to $T$ **do**
7:        $C \leftarrow E.\text{get\_state}()$
8:        $a_t \leftarrow \pi_\theta(s_{t-1})$
9:        $s_t, r_t \leftarrow E.\text{step}(a_t)$
10:        $F \leftarrow F \cup \{(s_{t-1}, a_t, r_t)\}$
11:     **end for**
12:     $\mathcal{S} \leftarrow \mathcal{S} \cup \text{CreateSegments}(F, L)$
13: **end for**
14: **for** $\sigma \in \mathcal{S}$ **do**
15:     $V_0 \leftarrow \frac{1}{|\mathcal{E}|} \sum_{e \in \mathcal{E}} V^e(\sigma_0)$
16:     $V_L \leftarrow \frac{1}{|\mathcal{E}|} \sum_{e \in \mathcal{E}} V^e(\sigma_L)$
17:     $R_\sigma \leftarrow \sum_{t=0}^{L-1} \gamma^t r_t$       ▷ Discounted reward sum
18:     $\mathcal{F}_{eval} \leftarrow 10 - \text{BinIndex}(R_\sigma, 10)$       ▷ Evaluative feedback
19: **end for**
20: $\mathcal{F}_{comp} \leftarrow \text{GetPreferencePairs}(\mathcal{S}, R_\sigma)$       ▷ Comparative feedback
21: **return** $\mathcal{F}_{eval}, \mathcal{F}_{comp}$

---

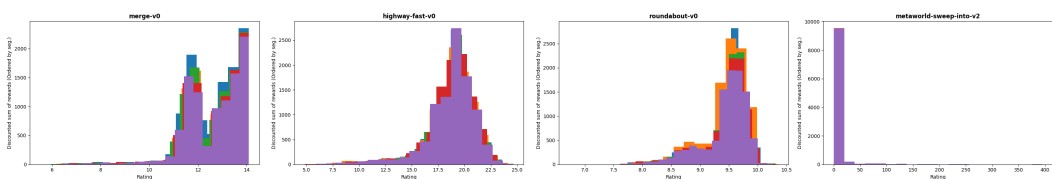

Figure 9: Cont. Histogram of total discounted reward distribution (for 10000 randomly sampled segments with max. length 50/ truncated at episode end). Each color represents a different generated feedback dataset.

Figure 10 shows the extracted ratings (plotted as 0-9) in relation to the underlying rewards.

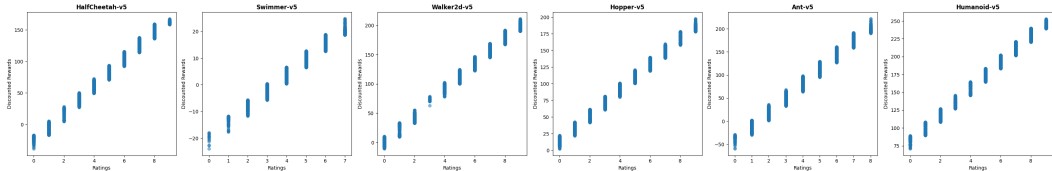

Figure 10: Comparison between ratings and total discounted reward (for 10000 randomly sampled segments with max. length 50/ truncated at episode end). Colors indicated the origin model of the rating (out of the expert model ensemble). Segments are sorted by discounted return.

**Comparative feedback** is directly extracted from the and therefore shares the characteristics with rating-based feedback. 1 contains the pseudo-code describing the feedback generation process for evaluative (rating) feedback, and comparisons.

Figure 11 and Figure 12 plot the return difference between preferred and non-preferred segments in the dataset. As the segments with higher return is preferred (in the optimal non-noise case), we only see values in the upper-left triangle. We see, that the segments pairs have a wide distribution of return

differences, i.e. the dataset contains pairs with very similar or dissimilar returns, i.e. very similar or dissimilar levels of performance with respect to the ground truth reward function. Furthermore, pairs are collected across a wide range of returns, indicating the effectiveness of our expert model-based segment sampling approach.

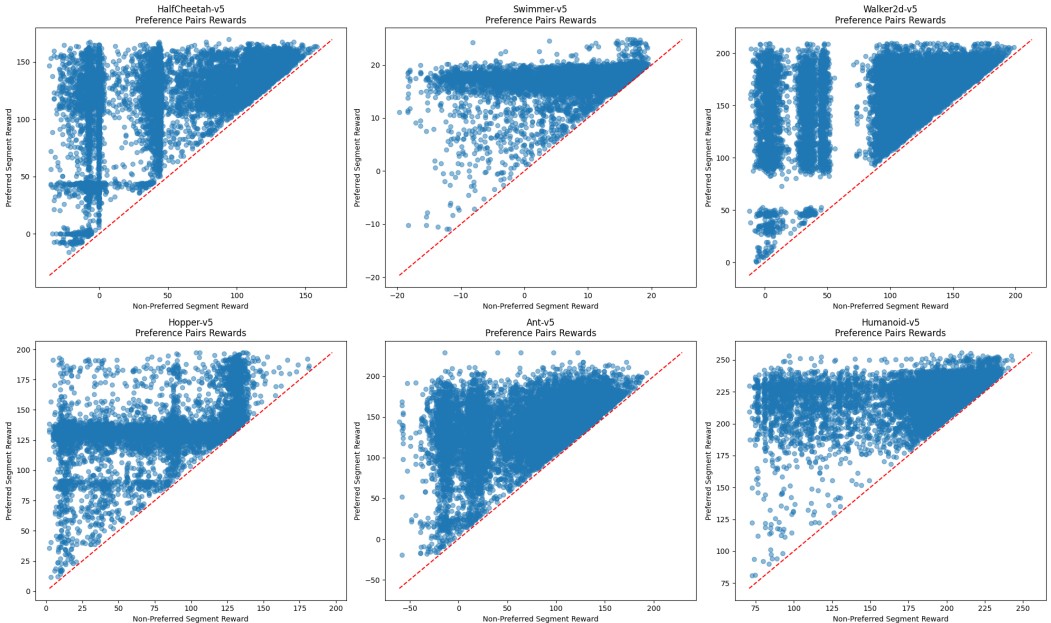

Figure 11: Scatter plot of discounted rewards for preference pairs: On the y-axis, we plot the rewards of the preferred segments, on the x-axis are the rewards of the non-preferred segments. Because the return of the preferred result in the optimal case (Noise 0.0) is always preferred, we only observe points above the diagonal.

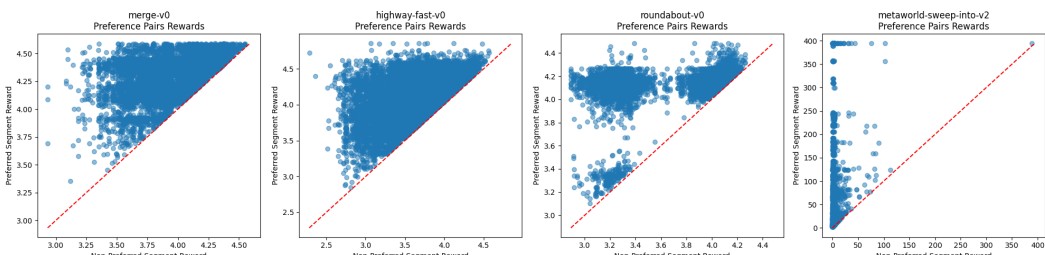

Figure 12: Continuation: Scatter plot of discounted rewards for preference pairs: On the y-axis, we plot the rewards of the preferred segments, on the x-axis are the rewards of the non-preferred segments. Because the return of the preferred result in the optimal case (Noise 0.0) is always preferred, we only observe points above the diagonal.

Figure 13 and Figure 14 show histograms of the segment pair return differences for different environments.

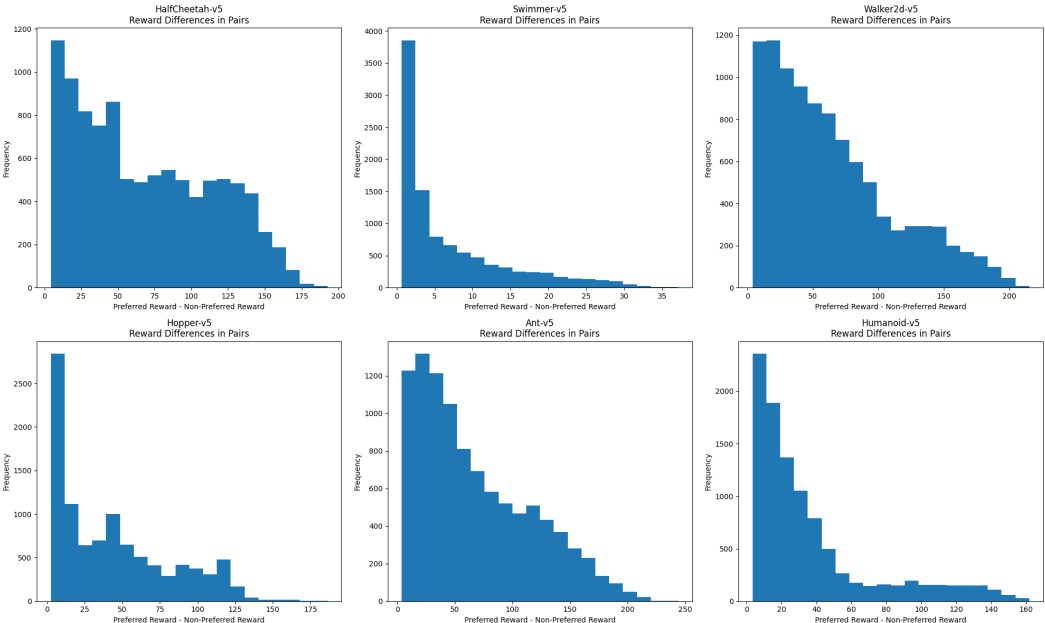

Figure 13: A histogram showing the distribution of return differences between preference pairs (for 10000 randomly sampled segments with max. length 50/ truncated at episode end).

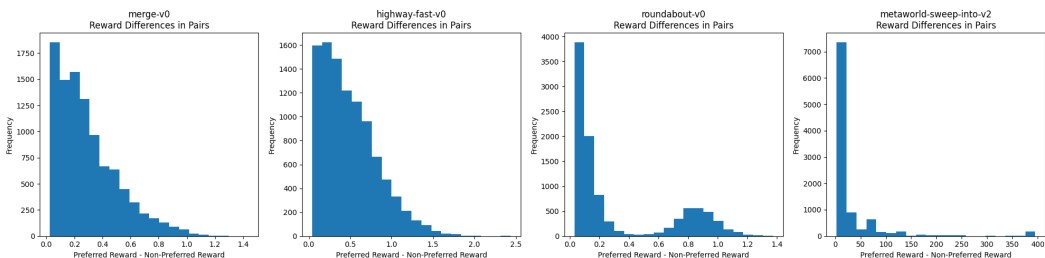

Figure 14: Continuation: A histogram showing the distribution of return differences between preference pairs (for 10000 randomly sampled segments with max. length 50/ truncated at episode end).

---

**Algorithm 2** Generation of Demonstrative and Corrective Feedback

---

**Require:** Environment $E$, Expert models $\mathcal{E} = \{e_1, \ldots, e_n\}$, State copies $\mathcal{C} = \{c_1, \ldots, c_m\}$, Segments $\mathcal{S} = \{s_1, \ldots, s_m\}$, Segment length $L$, Discount factor $\gamma$
**Ensure:** Demonstrative feedback $\mathcal{F}_{demo}$, Corrective feedback $\mathcal{F}_{corr}$

1: $\mathcal{F}_{demo} \leftarrow \varnothing, \mathcal{F}_{corr} \leftarrow \varnothing$
2: **for** $i \in \{1, \ldots, m\}$ **do**
3:      $D \leftarrow \varnothing$                                    ▷ Current demonstrations
4:      $R \leftarrow \varnothing$                                     ▷ Current expert returns
5:      **for** $e \in \mathcal{E}$ **do**
6:          $s \leftarrow E.\text{set\_state}(c_i)$
7:          $d \leftarrow \varnothing$                              ▷ Current demonstration
8:          **for** $j \in \{1, \ldots, L\}$ **do**
9:              $a \leftarrow e(s)$
10:            $s', r \leftarrow E.\text{step}(a)$
11:            $d \leftarrow d \cup \{(s, a, r)\}$
12:            $s \leftarrow s'$
13:          **end for**
14:          $D \leftarrow D \cup \{d\}$
15:          $R \leftarrow R \cup \{\sum_{t=0}^{L-1} \gamma^t r_t : (s_t, a_t, r_t) \in d\}$
16:      **end for**
17:      $k \leftarrow \arg\max R$
18:      $d^* \leftarrow D_k, r^* \leftarrow R_k$
19:      $r_{\text{orig}} \leftarrow \sum_{t=0}^{L-1} \gamma^t r_t : (s_t, a_t, r_t) \in s_i$
20:      **if** $r^* > r_{\text{orig}}$ **then**
21:          $\mathcal{F}_{demo} \leftarrow \mathcal{F}_{demo} \cup \{d^*\}$
22:          $\mathcal{F}_{corr} \leftarrow \mathcal{F}_{corr} \cup \{(s_i, d^*)\}$
23:      **end if**
24: **end for**
25: **return** $\mathcal{F}_{demo}, \mathcal{F}_{corr}$

---

## B.2   DISTRIBUTION AND EXAMPLES OF DEMONSTRATIVE AND CORRECTIVE FEEDBACK

A key novelty over existing work is the ability to also generate demonstrations/action advice and corrections dynamically for given segments based on direct access to the expert policy. 2 contains pseudo-code for the generation of demonstrative and corrective feedback. However, this approach raises several questions, which we should investigate: (1) How does the distribution of generated demonstrations/corrections look like? (2) How do they compare to the existing episodes within the dataset? (3) How does the choice of an expert policy influence the generated feedback?

**Comparison of Returns between Learner and Experts —** First, we want to make sure that the expert model policies that we use to *correct* and *demonstrate* advanced behavior, actually meaningfully improve over the baseline learner performance. As we can see in Figure 15 and Figure 16, expert policies can improve performance over the learner. However, depending on the situation, we see a wide range of achieved rewards for both policy types. This is caused by the initial start state of a trajectory, which might be highly non-optimal, and the expert policy might only be able to recover from a failure state but not achieve fully optimal reward.

**Examples for Learner and Corrected Trajectories —** To further illustrate the difference between weak *learner* trajectories and more optimal *expert* trajectories, we give some examples: Figure 17-Figure 18 highlight the differences in behavior by the weak and strong policy. As expected, we can identify that expert policies better exploit good initial states and are often able to recover from failure cases. More video examples are provided in the supplementary material.

## B.3   DETAILS OF DESCRIPTIVE FEEDBACK

In subsection 3.1, we have defined descriptive feedback as a mapping from abstract state-/action-feature values to a feedback value. We have implemented this feedback type via a clustering approach:

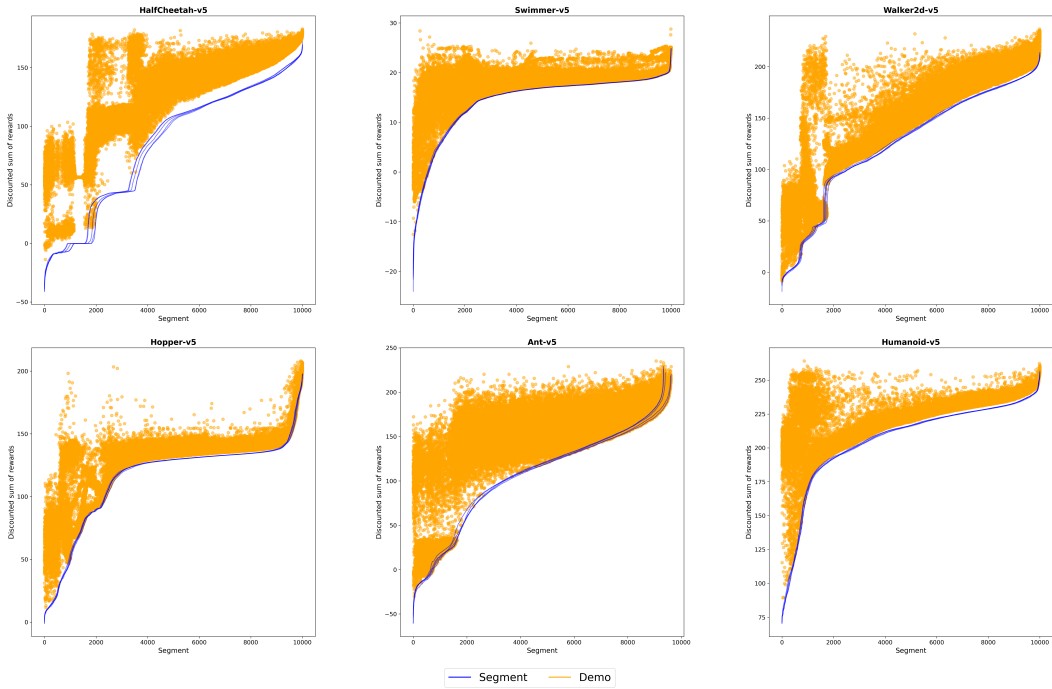

Figure 15: Comparison of achieved segment return between learner and expert demonstrations that we use as corrective and demonstrative feedback.

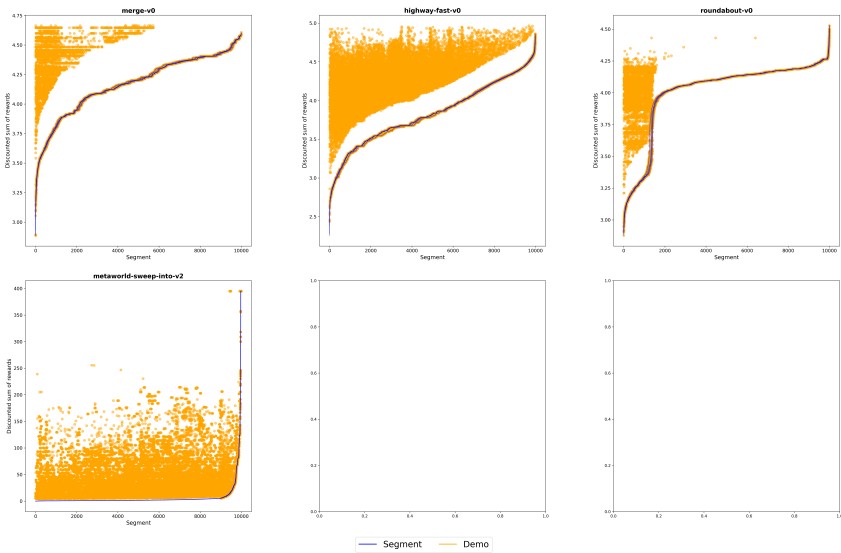

Figure 16: Comparison of achieved segment return between learner and expert demonstrations that we use as corrective and demonstrative feedback.

All collected state-action pairs $(s, a) \in \mathcal{D}$ are clustered into N clusters (with N being the number of feedback instances/queries). In our experiment, we match the number of clusters with the number of segments, e.g., $10.000$. We utilize mini-batch k-means clustering with a batch size of 1000 to cluster the $(s, a)$-pairs. We then average the observation, action, and optimality gaps for a cluster to get a single feedback instance $(s, a, v_{fb})$ that describes part of the state space without referring to a specific state.

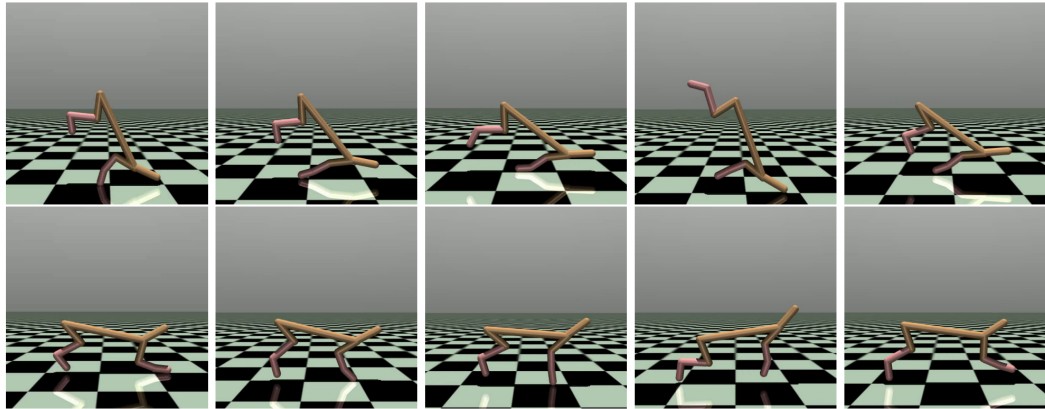

Figure 17: **HalfCheetah-v5** - Comparison of key-frames between learner and expert demonstrations, that we use as corrective and demonstrative feedback: Top Row are states from the *learner segment*, bottom row is the *expert segment*.

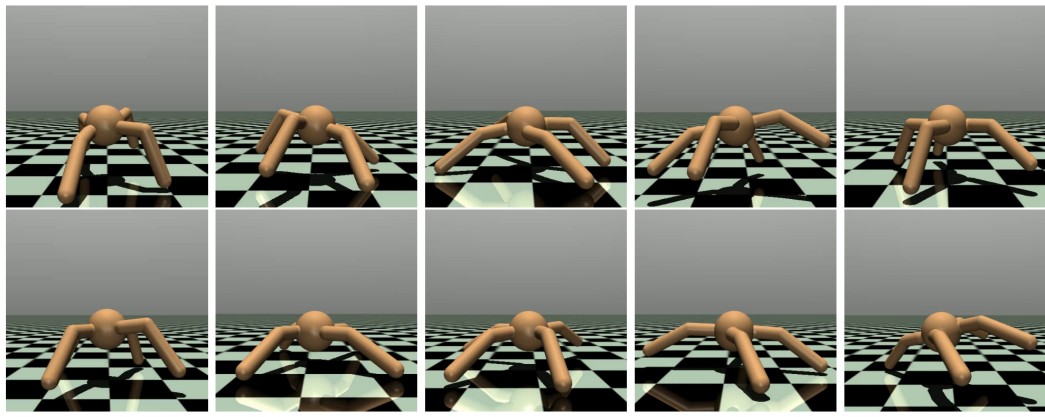

Figure 18: **Ant-v5** - Comparison of key-frames between learner and expert demonstrations that we use as corrective and demonstrative feedback: Top Row is stated from the *learner segment*, bottom row is the *expert segment*.

In Figure 19, we show a sub-sample of state-action pairs (5000 samples) alongside a sample of the computed averaged cluster representatives (500) and the associated step reward for each pair marked in red. In Figure 20, we show t-sne projections of the full set clusters alongside the associated descriptive rewards

### B.4 Limitations and Alternative Formulations

Lastly, the generation scheme has some limitations. The biggest limitation in our mind, is the potential mismatch between optimality of the evaluative/descriptive feedback, which is based on the underlying reward function, and the demonstrative/corrective feedback that is based on expert policies. While these experts policies were optimized with respect to the equivalent ground-truth discounted reward functions, they might still be sub-optimal compared to an optimal policy. However, in none of our explored environments, we have access to an optimal expert policy.

The feedback datasets were collected over trajectories sampled by rolling out the models from various checkpoints of the expert policies. This can be seen both as a positive or negative thing: On the positive thing, this ensures a better comparability between feedback types (including demonstrative/corrective feedback(, because this means feedback is available for roughly the same space of trajectories across types, and should therefore transport comparable underlying biases and non-optimality. On the flip-side, this means that the learned reward functions are only well-defined for the trajectories in the

---

**Algorithm 3** Generation of Descriptive Feedback

---

**Require:** Segments $\mathcal{S} = \{s_1, \ldots, s_n\}$, Number of feedback instances $k$
**Ensure:** Descriptive feedback $\mathcal{F}_{desc}$, Descriptive preferences $\mathcal{F}_{desc.pref.}$

1: $\mathcal{X} \leftarrow \varnothing, \mathcal{R} \leftarrow \varnothing$
2: **for** $s \in \mathcal{S}$ **do**
3:     $\mathcal{X} \leftarrow \mathcal{X} \cup \{Concatenate(x_i, a_i) : (x_i, a_i, r_i) \in s\}$
4:     $\mathcal{R} \leftarrow \mathcal{R} \cup \{r_i : (x_i, a_i, r_i) \in s\}$
5: **end for**
6: $C \leftarrow \text{KMeans}(\mathcal{X}, k)$                                              ▷ Cluster assignments
7: $\bar{\mathcal{X}} \leftarrow \{\frac{1}{|C_i|} \sum_{j \in C_i} \mathcal{X}_j : i \in \{1, \ldots, k\}\}$         ▷ Cluster representatives
8: $\bar{\mathcal{R}} \leftarrow \{\frac{1}{|C_i|} \sum_{j \in C_i} \mathcal{R}_j : i \in \{1, \ldots, k\}\}$        ▷ Cluster mean rewards
9: $\mathcal{F}_{desc} \leftarrow \{(\bar{\mathcal{X}}_i, \bar{\mathcal{R}}_i) : i \in \{1, \ldots, k\}\}$           ▷ Cluster descriptions
10: $\mathcal{F}_{desc.pref.} \leftarrow \text{GetPreferencePairs}(\bar{\mathcal{X}}, \bar{\mathcal{R}})$
11: **return** $\mathcal{F}_{desc}, \mathcal{F}_{desc.pref.}$

---

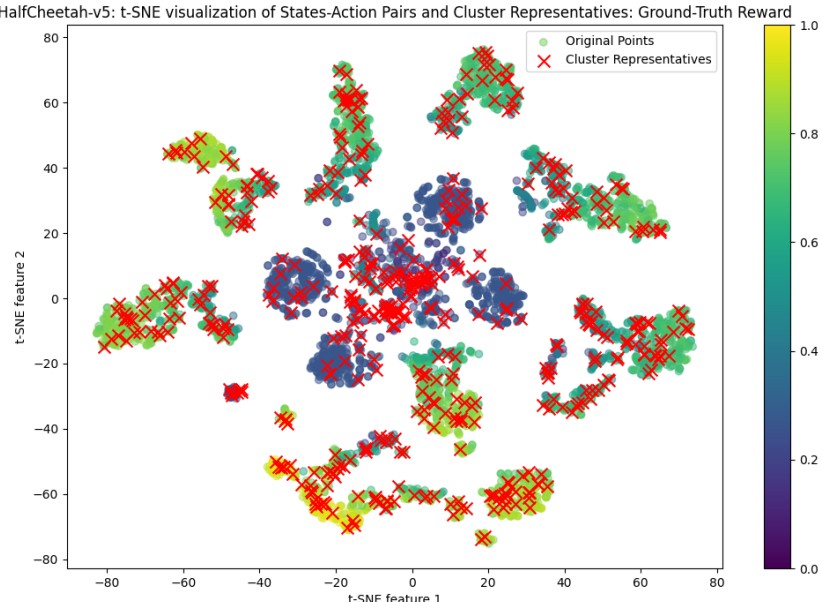

Figure 19: t-SNE projection of state-action pairs, colored with the associated ground-truth reward. Overlayed, marked by $X$ are computed cluster representatives. These cluster representatives, alongside the average reward of underlying pairs, are used as descriptive feedback.

datasets, i.e. also better-than-expert trajectories that deviate significantly from the observed behavior, must be considered out-of-distribution. We therefore do not expect that trained RL models can easily surpass expert performance. Indeed, this is mostly supported by the experimental results.

Our method is well suited for generating fixed, offline training datasets. However, our approach requires some modification to generate online oracle feedback. This includes initializing the histogram of ratings based on an initial dataset, but updating it based on online data. Similarly, the clustering used to generate descriptive feedback must also be re-computed on a dynamic buffer to adapt to online trajectories. In the provided code repository, we provide an implementation of dynamic feedback oracle which can be used for online RLHF with simulated multi-type feedback.

**Alternative Formulations: —** During the design process, we discussed several alternative ways to

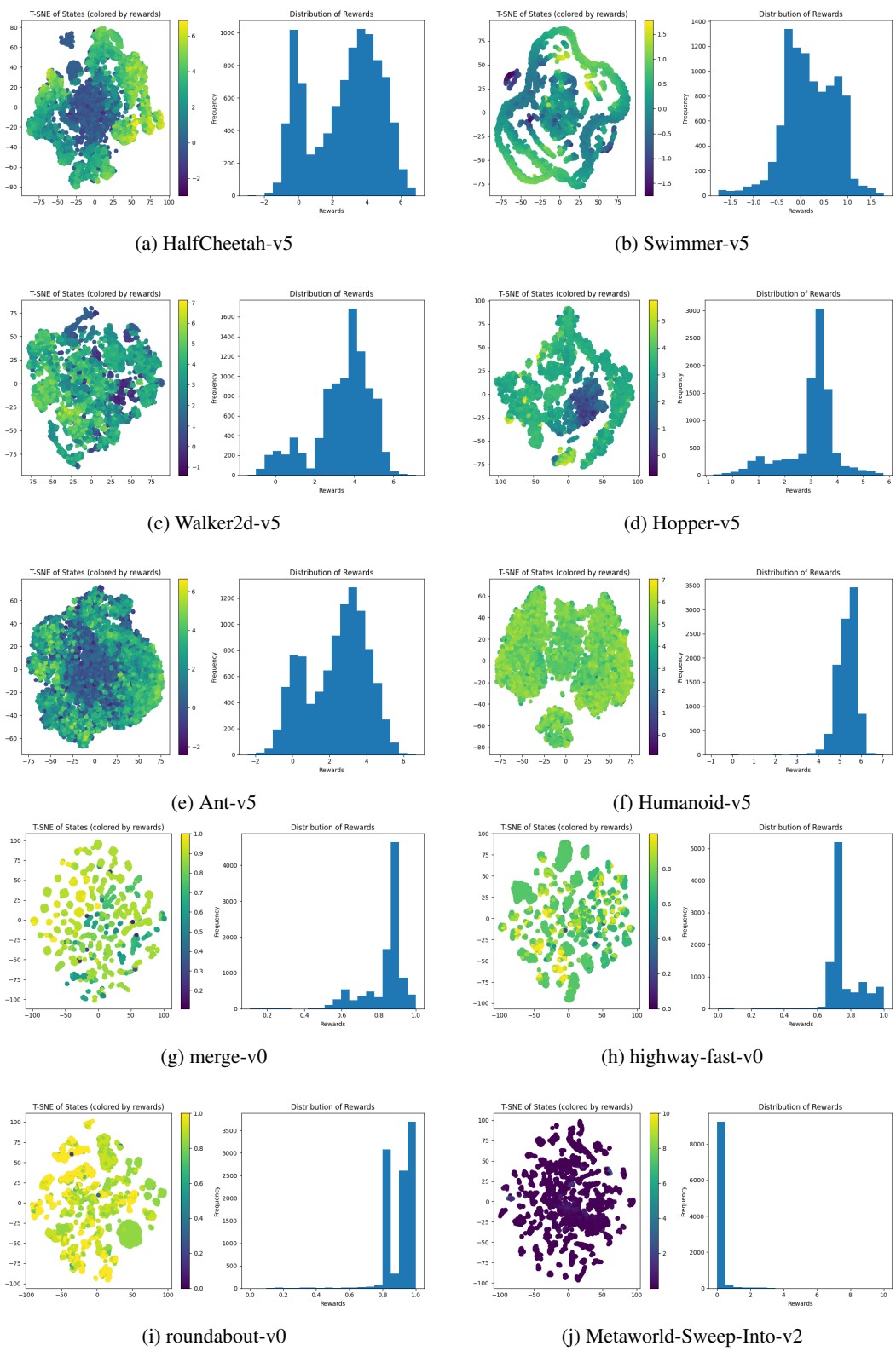

Figure 20: Descriptive Feedback Analysis for the benchmarked environments. Each subfigure shows the T-SNE visualization of states (left) and the distribution of rewards (right) for the respective environment.

generate synthetic feedback:

- (Evaluative feedback) Disagreement with expert policy: A possible alternative to using the expert's value function is to base the feedback directly on the policy. Suppose the selected action observed in the segment deviates from the agent's prediction. In that case, we assume it is against the intended series of actions that the expert agent prefers. *However, this formulation does not have room for alternative solution strategies that might be recognized by a human annotator, including segments that are even better than the expert.* In both cases, the optimality-gap-based approach will lead to low and potentially even negative gaps, which will be converted into high-score feedback.

- (Descriptive feedback) The formulation of descriptive feedback is more undetermined than the other types. We explored an alternative approach to assign scores or preferences to features instead of segments using feature attribution methods of the value estimator. This allows us to identify features that are relevant for a high/low value. We can then interpret the difference in prediction for the presence/absence of an important feature as a reward assignment. However, we have found the attributions generated by methods such as *Integrated Gradients* (Sundararajan et al., 2017) overly noisy to be effective in our context. However, in our experiments, the attributions maps proved rather unreliable and only provided a weak signal due to being similar across states. Furthermore, they were only suited to create description of state, and not state-action pairs.

An additional approach based on regret, that we explored more extensively, is presented in the following.

### B.5 ALTERNATIVE GENERATION OF EVALUATIVE AND COMPARATIVE FEEDBACK WITH OPTIMALITY GAPS

We extensively explored to use the value function of an expert model as a source of feedback. Existing work (Brown et al., 2019a; Xue et al., 2023; Christiano et al., 2017) utilizing simulated feedback (often called an *oracle*), use the ground-truth reward function of the environment to simulate human feedback.

In actor-critic RL algorithms like SAC (Haarnoja et al., 2018) or PPO (Schulman et al., 2017), we can utilize the value $V(s) \in \mathbb{R}$ or q-value $Q(s, a) \in \mathbb{R}$ estimate directly to generate feedback values, and the learned policy for demonstrative feedback. Similarly, we can directly transfer this approach to purely value-based RL like Q-Learning.

We explored an alternative way to generate **evaluative feedback** on a notion similar to *regret*, i.e., we rate a segment not based on the actual collected environment reward but instead on whether it showcases optimal behavior from the perspective of an expert model. This notion aligns more closely with reward behavior observed in humans, which have been shown to rate *relative to their expectations* (MacGlashan et al., 2017; Knox et al., 2022; Jeon et al., 2020). Furthermore, feedback based on ground-truth rewards is not entirely suited for sparse-reward tasks, as we might only observe a few rewards during a typical segment, resulting in a weak learning signal. The observed regret is computed by comparing the expected future rewards with the observed ones:

$$\Delta_{opt}(\xi_{0:H}) := V_e(s_0) - \left( \sum_{i=0}^{H-1} \gamma^i r_i + \gamma^H V_e(s_H) \right)$$

**Regret and Optimality Gap** — We want to briefly clarify the relation between the formal **regret** as commonly defined in RL research and our definition of

**Lemma B.1.** *Equality of Regret and Optimality Gap  Under the assumption that the expert policy is optimal, the optimality gap $\Delta_{e,opt}$ is equivalent to the expected regret for a segment in **fixed-horizon tasks**.*

*Proof.* We can infer this from the common definition of regret $R_H$ over a time horizon $H$ (in our case, the length of a segment). With ground-truth reward function $r(s_t, a_t)$ and $(s_t, a_t)_{t=0}^{H}$ being the

segment:

$$R_H := \mathbb{E}[\sum_{t=0}^{H} r(s_t, a_t^*) - r(s_t, a_t)]$$

$$= \mathbb{E}[\sum_{t=0}^{H} r(s_t, a_t^*) - \sum_{t=0}^{H} r(s_t, a_t)]$$

$$= \mathbb{E}[(\sum_{t=0}^{H} r(s_t, a_t^*) + \sum_{t=H+1}^{T} r(s_t, a_t^*) - \sum_{t=H+1}^{T} R(s_t, a_t^*)) - \sum_{t=0}^{H} r(s_t, a_t)]$$

$$= \mathbb{E}[(\sum_{t=0}^{T} r(s_t, a_t^*) - \sum_{t=H+1}^{T} r(s_t, a_t^*)) - \sum_{t=0}^{H} R(s_t, a_t)]$$

$$= \mathbb{E}[V^*(s_0) - V^*(s_H) - \sum_{t=0}^{H} r(s_t, a_t)]$$

$$\overset{V*\equiv V_e}{=} \mathbb{E}[V_e(s_0) - V_e(s_H) - \sum_{t=0}^{H} r(s_t, a_t)]$$

$\square$

However, in infinite horizon tasks, i.e., *Mujoco* is treated as an infinite horizon task in the common implementations of PPO, we often observe that $V^e(s_0)$ and $V^e(s_H)$ are very similar or even equivalent, except for states that show catastrophic failure, because states cannot be easily differentiated from each other and therefore will have very similar value estimates. Theorem B.1 also does not hold for the infinite horizon case due to infinite sums.
In practice, this means that the *optimality gap* $\Delta_{opt,e}$ is similar to the negative sum of (discounted) ground truth rewards, i.e., it reduces to the standard implementations of feedback.

Another complicating factor is environmental stochasticity. Due to these factors, we chose the term *optimality gap* instead of *regret* to indicate the distinction.

**Comparison with Environment Ground-Truth Feedback —** As described above, we use regret-based optimality gap w.r.t., an expert value function, as our foundation for evaluative feedback. As a first sanity check for the validity of this generated feedback, we investigate the relationship between these optimality gaps and sums of ground-truth reward (akin to existing work): Visible is a linear correlation between achieved rewards and the negative optimality gap. In particular for non-episodic tasks with low influence of stochasticity like *Mujoco*-environments, negative optimality gaps, and discounted total rewards almost co-inside, with the imperfect nature of value estimates leading to the introduction of some noise. For some of the other environments, we observe both the difficulty and promise of our approach over ground-truth reward: Return (i.e., the sum of (discounted) rewards over a segment) is an uninformative measure in many environments with sparse rewards (like *Atari*). In contrast, the simulated reward can serve as a stronger signal.

**The effect of value-function ensembles on simulated feedback —** In the main paper, we use the ground truth reward function to generate feedback. We have experimented with using multiple trained agents to get more reliable value estimates, including a broader coverage and the ability to detect out-of-distribution samples that may lead to faulty results. We observed that some models tend to slightly over- or under-estimate the return, leading to slightly overlapping bins. We argue that this expert model noise is compatible with the expected behavior of human labels. Such label noise has been reported in crowd-sourcing tasks by human annotators. We average the optimality gap predictions across four models to achieve representative results.

We see that the value estimates for austere environments are often close together, and the averaged estimates have less noise, which means the estimated optimality gaps more closely mirror the observed ground-truth returns. For more complex environments, estimates can vary more. In the future, we might want to investigate further increasing the number of expert value functions for even more reliable optimality gap estimates.

## B.6 EXPERT POLICY: RL HYPERPARAMETER SETTINGS

To train expert models, we chose the default hyperparameter settings of RL training given in *Stable-Baselines3 Zoo* (Raffin, 2020).

Table 4: RL-Training hyperparameters for *Mujoco* expert agents trained with PPO.

| Hyperparameters for Mujoco Experiments: PPO | | | |
|---|---|---|---|
| Environment | Ant-v5 | Swimmer-v5 | Walker2d-v5 |
| Algorithm | SAC | | |
| Network Arch. | 3-layer MLP | | |
| Hidden size | 256 | 256 | 64 |
| Training Timesteps | 1e6 | 1e6 | 1e6 |
| Num. Parallel Env. | 1 | 4 | 1 |
| Batch Size | 64 | 256 | 32 |
| Learning Rate | 2.063e-5 | 6e-4 | 5e-05 |
| Num. Epochs | 20 | 10 | 20 |
| Sampled Steps per Env. | 512 | 1024 | 512 |
| GAE lambda | 0.92 | 0.98 | 0.95 |
| Gamma | 0.98 | 0.9999 | 0.99 |
| Ent. Coeff | 0.0004 | 0.0554757 | 0.0005 |
| VF Coeff. | 0.581 | 0.38782 | 0.872 |
| Max. Grad Norm | 0.8 | 0.6 | 1.0 |
| Clip Range | 0.1 | 0.3 | 0.1 |
| Obs. Normalization | false | | |
| Reward Normalization | false | | |

Table 5: RL-Training hyperparameters for *Mujoco* expert agents trained with SAC.

| Hyperparameters for Mujoco Experiments: SAC | | | |
|---|---|---|---|
| Environment | Ant-v5 | Hopper-v5 | Humanoid-v5 |
| Algorithm | | SAC | |
| Network Arch. | | 3-layer MLP | |
| Hidden size | 256 | 256 | 256 |
| Training Timesteps | 1e6 | 1e6 | 2e6 |
| Num. Parallel Env. | 1 | 1 | 1 |
| Batch Size | 256 | 256 | 256 |
| Learning Rate | 3e-4 | 3e-4 | 3e-4 |
| Gamma | 0.99 | 0.99 | 0.99 |
| Ent. Coeff | "auto" | "auto" | "auto" |
| Obs. Normalization | false | | |
| Reward Normalization | false | | |

Finally, in Table 8, we report the final evaluation returns by the trained expert policies (no-deterministic sampling, ten episodes). We ran five separate seeds for each environment. We selected the four best-performing seeds as part of the model ensemble for simulated feedback.

Table 6: RL-Training hyperparameters for *Highway-Env* expert agents trained with PPO.

| Hyperparameters for Highway-Env: PPO | | | |
|---|---|---|---|
| Environment | merge-v0 | highway-fast-v5 | roundabout-v5 |
| Algorithm | PPO | | |
| Network Arch. | 3-layer MLP | | |
| Hidden size | 256 | 256 | 256 |
| Training Timesteps | 1e5 | 1e5 | 2e5 |
| Num. Parallel Env. | 64 | 64 | 64 |
| Batch Size | 32 | 32 | 32 |
| Learning Rate | 5e-4 | 5e-4 | 5e-4 |
| Num. Epochs | 10 | 10 | 10 |
| Gamma | 0.8 | 0.8 | 0.8 |
| Ent. Coeff | 0.0 | 0.0 | 0.0 |
| Obs. Normalization | false | | |
| Reward Normalization | false | | |

Table 7: RL-Training hyperparameters for *Metaworld* expert agents trained with SAC.

| Hyperparameters for Metaworld Experiments: SAC | |
|---|---|
| Environment | Ant-v5 |
| Algorithm | SAC |
| Network Arch. | 3-layer MLP |
| Hidden size | 256 |
| Training Timesteps | 1e6 |
| Num. Parallel Env. | 1 |
| Batch Size | 256 |
| Learning Rate | 3e-4 |
| Gamma | 0.99 |
| Ent. Coeff | "auto" |
| Obs. Normalization | false |
| Reward Normalization | false |

Table 8: Sorted expert evaluation scores for selected environments, with averages. We only retain the top four expert models to generate synthetic feedback.

| HalfCheetah-v5 | Walker2d-v5 | Swimmer-v5 | Ant-v5 | Hopper-v5 | Humanoid-v5 |
|---|---|---|---|---|---|
| **5549.8062** | **5860.0417** | **359.1049** | **4306.0373** | **3556.5895** | **6254.5279** |
| **5254.2713** | **5071.9133** | **355.3227** | **4132.0158** | **3268.0585** | **5893.5920** |
| **5124.0117** | **4018.3773** | **320.6619** | **4122.3491** | **3250.1369** | **5853.7701** |
| **5095.0633** | **3717.2990** | **317.2805** | **3989.7485** | **3205.6200** | **5682.9198** |
| 4907.9425 | 3098.4732 | 295.3936 | 3601.7154 | 3191.8214 | 5403.7185 |
| 5186.2190 | 4353.2209 | 329.5527 | 4030.3732 | 3294.4453 | 5817.7057 |

## B.7 VISUALIZING THE ARTIFICIAL NOISE IN GENERATED FEEDBACK

Finally, we want to visualize the effect of introduced noise on the generated feedback. Figure 21 and Figure 21 shows the effect of truncated noise added onto the reward distribution. Figure 24 shows the impact of noise on the descriptive feedback. We showcase the adaptations for one of the datasets in the *HalfCheetah-v5* environment.

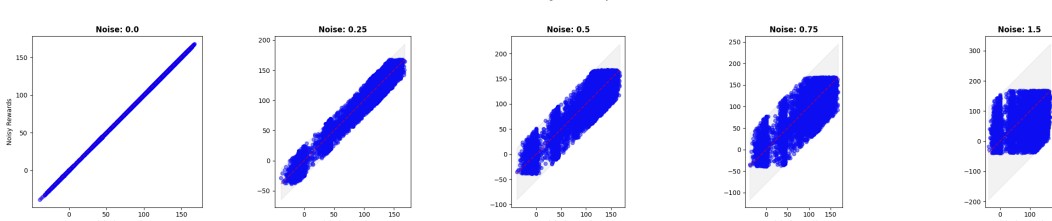

Figure 21: Scatterplot displaying original and perturbed rewards: With increasing noise, the underlying reward distribution gets perturbed.

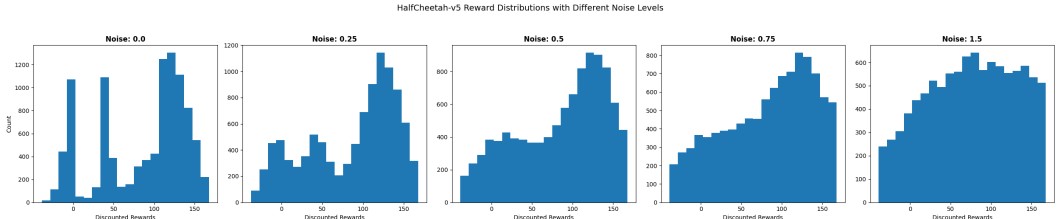

Figure 22: Histogram displaying original and perturbed rewards: With increasing noise, the underlying reward distribution gets perturbed.

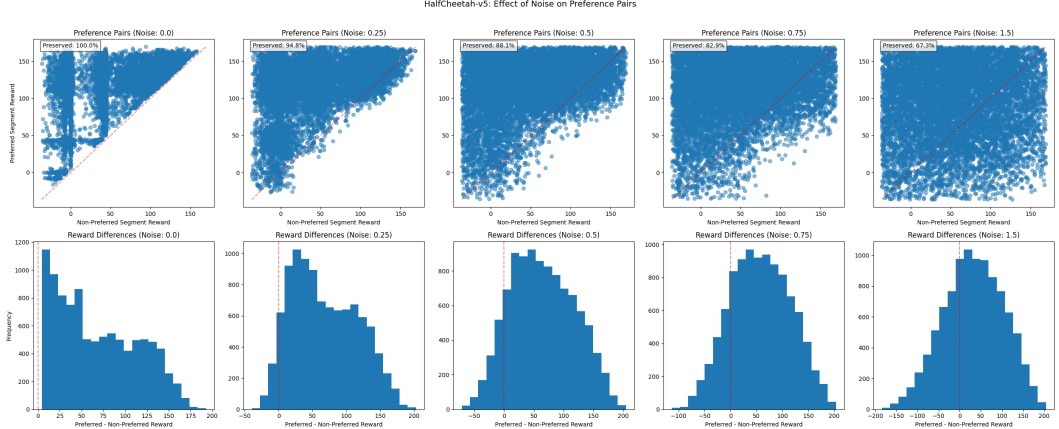

Figure 23: Scatterplot displaying the rewards of **preference pairs**: For optimal feedback, pairs always have a perfect relationship between good and bad reward. For increased noise, more and more pairs get flipped, i.e., indicate a wrong preference w.r.t. the ground-truth reward function.

Our library includes utilities to investigate and visualize the details of the datasets, including the perturbed data. We argue that this kind of transparency is crucial for reproducibility.

As a future extension of our work, we may develop common reporting standards for the content of feedback datasets, including sample and rating distributions, as well as underlying assumptions.

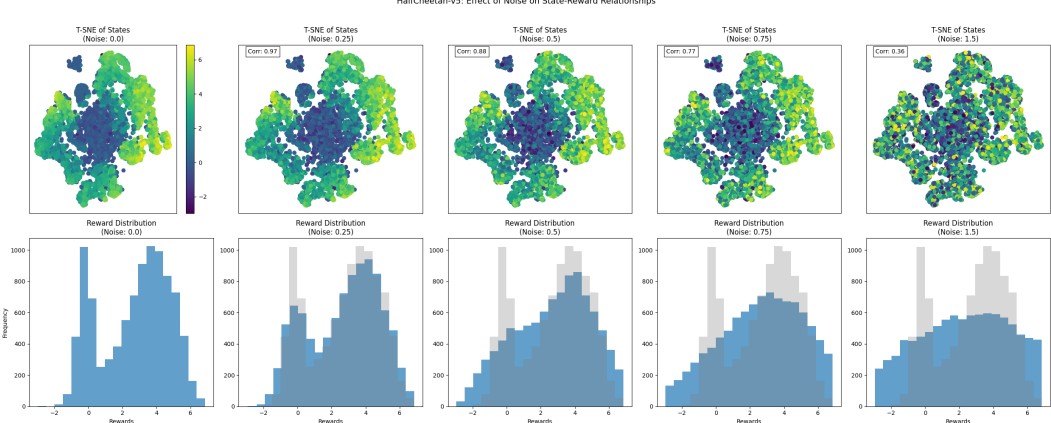

Figure 24: Displaying the effect of artificial noise on **descriptive feedback**: The assigned cluster reward values are perturbed by the introduced noise.

## C  EXPERIMENTS WITH SIMPLIFIED REWARDS

In preparation for the main experiments reported in this paper, we ran trials using a method that used the reward predictions for each step separately. In this setup, we adopted reward models trained from five types (evaluative, comparative, descriptive, corrective, and demonstrative) of synthetically generated feedback to provide dense rewards to an agent trained using the SAC algorithm in the Half Cheetah v3 environment. To train the reward models, we employed a slightly modified version of the training procedure described earlier, where we used feedback given on single steps as inputs to the neural network that was trained to predict a reward for that step. In the case of the runs using ensemble networks, the individual model outputs were averaged to get the final reward for the current step.

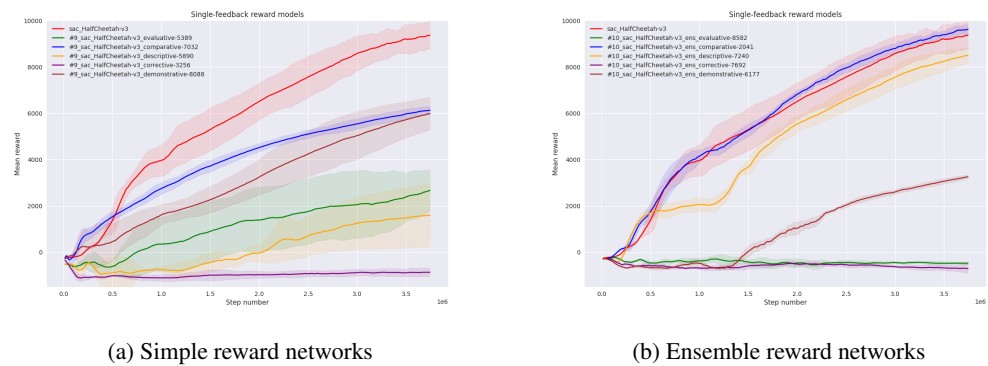

(a) Simple reward networks          (b) Ensemble reward networks

Figure 25: RL Training Curves for training from different individual feedback reward models (displayed are the ground-truth rewards). The line labeled Expert represents the performance of the expert policies used for generating feedback.

### C.1  USING REWARD FUNCTION ENSEMBLES

The summary of mean reward curves from 4 training runs is shown in figure 25. First, we compare the training of the simple reward model networks (left) to those trained using ensembles (right). In these figures, we can see that, in some cases, using ensembles dramatically improves performance. For example, after introducing ensembles to the comparative and descriptive feedback models, their performance reached that of the agent trained using environment rewards (labeled *Expert*), and the standard deviation of the rewards achieved during the four runs also decreased. In other cases, such as for evaluative, demonstrative, and corrective feedback models, the performance of the trained RL

agent decreased.

## C.2 COMBINATION OF FEEDBACK TYPES BY AVERAGING

In addition to using the feedback models by themselves, we can combine them by averaging their reward predictions. Additionally, since the magnitudes of these outputs are not on the same scale and can vary, we also need to normalize them before averaging. We keep a rolling mean and standard deviation for this and update them using Welford's algorithm (Welford, 1962).

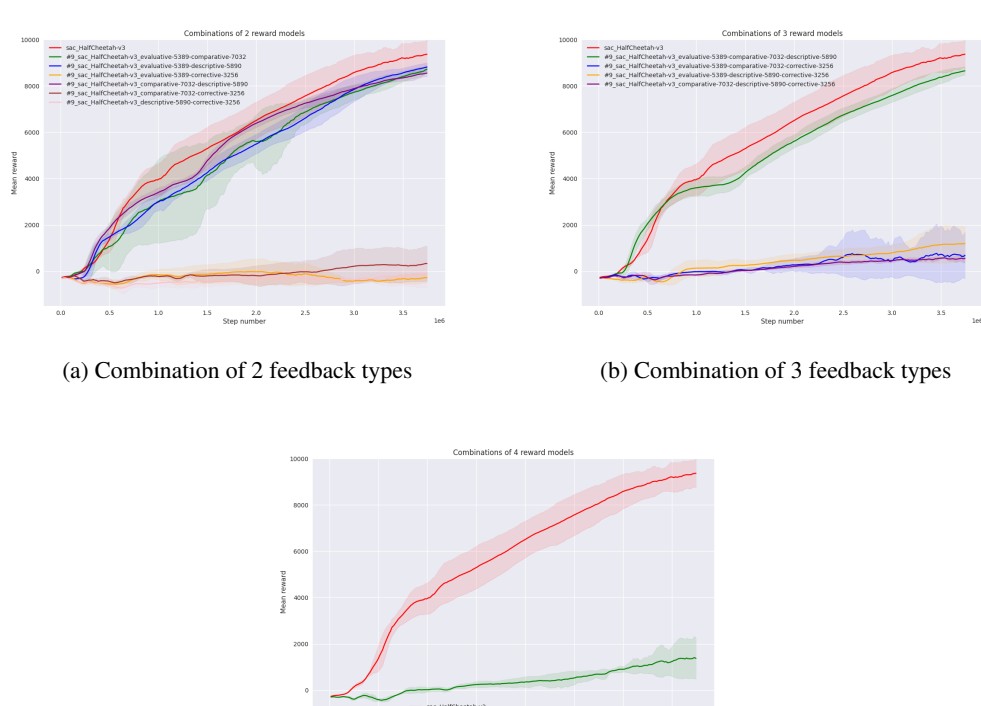

(a) Combination of 2 feedback types          (b) Combination of 3 feedback types

(c) Combination of 4 feedback types

Figure 26: RL Training Curves for training from different combined feedback reward models (displayed are the ground-truth rewards). The line labeled as *sac_halfCheetah-v3* represents the performance of the expert policies used for generating feedback.

The results of runs using different numbers of feedback types are in figure 26. From the training sessions using the combination of 2 feedback types, we can see that some significantly improved the performance over agents trained using their feedback components, reaching mean rewards similar to what the expert model obtained while also improving the standard deviation of rewards throughout the different runs. Specifically, all combinations that did not include the corrective feedback type achieved better scores than both of their single feedback components.

We get a similar picture by looking at the figures of combinations of 3 feedback types, where if we do not include the worst-performing corrective feedback, then the rewards obtained by the trained RL agent significantly increase. This might also be the reason for the relatively poor performance of the agent trained with four feedback types, including the corrective feedback.

Overall, combining multiple modalities of feedback, the agents' scores were higher than those obtained using the worst single feedback component - but in many cases, better than all - which indicates that the strengths of these individual modalities are combined through mixed-feedback training.

# D   REWARD MODEL TRAINING DETAILS

In this appendix, we want to discuss details of reward model training, including ablations and hyperparameters settings.

## D.1   TRAINING DETAILS

Rewards were trained on the collected feedback datasets in a standard, supervised way. For the Mujoco environments, we used a 6-layer MLP with 256 hidden units and ReLu activation function. We use the *Adam* optimizer with a learning rate of $1e-5$ and weight decay $1e-2$. Models are trained for 100 epochs, with early stopping (5 patience epochs) enabled. We use a batch size of 128.

Based on best practices from previous work and our preliminary experiments (see subsection C.1), we use reward function ensembles instead of single monolithic models and average the predictions of the submodels. We implemented these ensembles by using *Masksembles* layers (Durasov et al., 2021) after each fully-connected layer of the network. Similar to techniques like Monte-Carlo Dropout, a model with *Masksembles* computes multiple predictions of partially overlapping models during inference. We chose a mask number of 4 with a scale factor of 1.8, corresponding to four ensemble members. Using Masksembles reduced the required training compute and parameter count compared to a "true" reward model ensemble while achieving comparable performance in many cases (Durasov et al., 2021; Bykovets et al., 2022).

## D.2   LEARNING FROM NOISY FEEDBACK

For our baseline experiments, we have analyzed reward modeling and RL performance for optimal feedback (w.r.t. to the expert models) (Griffith et al., 2013). However, expecting real human feedback to be optimal is a very strong assumption that can generally not be met. We investigated the performance of different feedback types under increasing noise to the generated labels. In order to enable meaningful comparison between different feedback types, we designed a consistent perturbation approach across feedback types:

**Basic approach: —** We modify the underlying reward distribution by adding Gaussian noise of varying degree. The values for evaluative feedback and cluster description reward are updated accordingly, and preferences are flipped if the rewards change. The level of noise is controlled by single parameter $\beta$. First, the range of the reward distribution is computed, e.g, $[r_{min}, r_{max}]$. For evaluative feedback and corrections, this is the range of discounted segment returns; for descriptive feedback this coincides with the range of single cluster rewards. Each single feedback value is then perturbed by additive noise sampled from a truncated gaussian distribution with $\mu = v_{fb}$ and $\sigma = \beta * |r_{max} - r_{min}|$.

**Truncated Gaussian Distribution: —** We use a truncated distribution, to ensure comparability with the binning based approach to generate rating feedback, as well as to respect bounds of observations/actions, which is relevant for demonstrative feedback. Compared to a Gaussian distribution with fixed clipping values, sampling from truncated Gaussian distribution has a lower variance: Based on a brief analysis of both approaches on a subset of our data, we expect that we need to set the standard deviation $sigma$ of the truncated normal distribution $\mathcal{N}_T$ to around four times the standard deviation of an un-truncated normal distribution $\mathcal{N}$ to have a comparable level of perturbation (as measured by the $L2$-norm between original and perturbed value distribution). The $\beta$-values reported in this paper are for the truncated normal distribution, thus for replicating the results with non-truncated gaussian distribution for noise would therefore correspond roughly to a value $\frac{\beta}{4}$.

**Rating Feedback —** We perturb rating feedback, by adding from truncated Gaussian distribution with mean at the original rating, and the standard deviation controlled by the parameter $\beta \in \mathbb{R}^+$, and truncate the values at the edges of the rating scale. This simulates the effect of a human user "incorrectly" switching between two neighboring ratings.

**Comparative, Corrective, Descriptive Preferences —** For these preference-based feedback types, we assign labels according to the newly perturbed segment return distribution. In effect, this means that preference pairs with similar performance are more often flipped than examples with a clear difference in performance, which matches our intuition. However, this approach might actually underestimate human error, that can be observed in real labeling tasks due to miss-clicks or miss-

understanding.

**Demonstrative Feedback —** Because we model demonstrative feedback effectively as preference-based feedback, we could also employ label flipping here. However, this does not represent the type of error a human labeler would make. Instead, for demonstrative feedback, noise should occur in sub-optimal demonstrations, i.e., sub-optimal action selection. For the sake of simplicity, being able to add distortion to already collected data, we do *not perform full rollouts* with a distorted policy. Instead we apply additive Gaussian noise on the observations and actions, simulating imperfect demonstrations. We acknowledge that this is a simplistic assumption and will consider further investigations and encourage further investigations to enable more human-aligned perturbations.

**Descriptive Feedback —** As descriptive feedback maps to scalar rewards, similar to rating feedback, we use a uniform noise schema for rating feedback. Because there are now fixed bounds for ratings (i.e., 1-10), we infer the feedback range based on the collected segments within the datasets $\mathcal{D}$ and single-step rewards. Feedback scores are then modified by uniform sampling from a truncated normal distribution with width $\beta * |max_{\mathcal{D}}(r) - min_{\mathcal{D}}(r)|$, conforming to the existing range of $\beta \in \mathbb{R}^+$.

To summarize, we can introduce different noise levels to the different feedback types, controlled by a parameter $\beta$, corresponding to a comparable degree of perturbation. For our experiments, we investigate the behavior of reward functions within a range of $\beta$ from 0.1 to 3.0, A $\beta$-value of 3.0 already corresponds to a very severe change in distribution.

### D.3 REWARD MODEL TRAINING RESULTS

Shown in Figure 27 and Figure 28, almost all reward models train effectively with optimal feedback, achieving low validation loss. We conclude that the reward models can fit the reward data to a high degree, and in turn that our chosen architecture and training protocol are generally appropriate for the task.

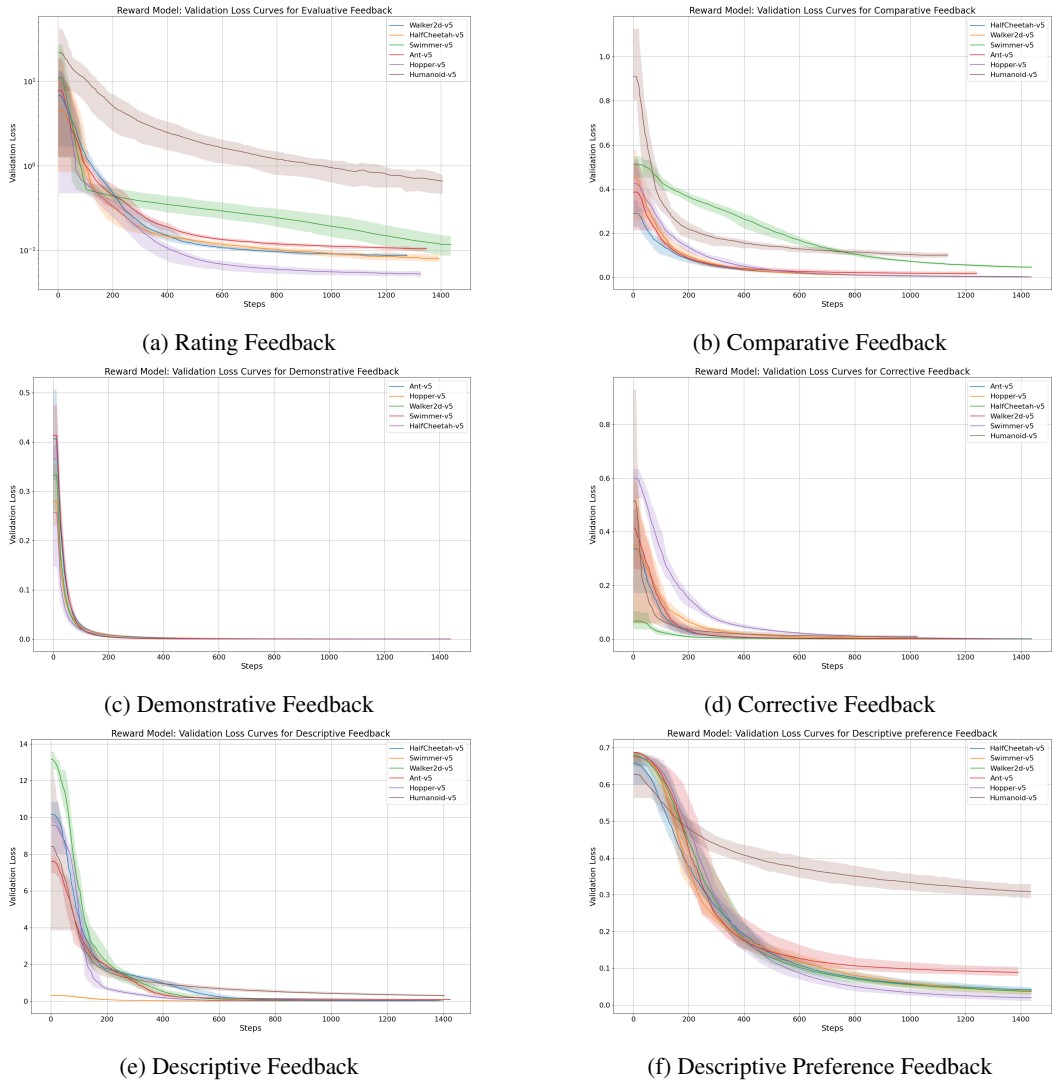

(a) Rating Feedback

(b) Comparative Feedback

(c) Demonstrative Feedback

(d) Corrective Feedback

(e) Descriptive Feedback

(f) Descriptive Preference Feedback

Figure 27: Validation Loss Curves for Different Feedback Types (**Mujoco**): Steps is the number of optimizer steps. Averaged over five feedback datasets.

When analyzing results for *MetaWorld* and *Highway*, we observe more variance compared to the Mujoco-Envs.

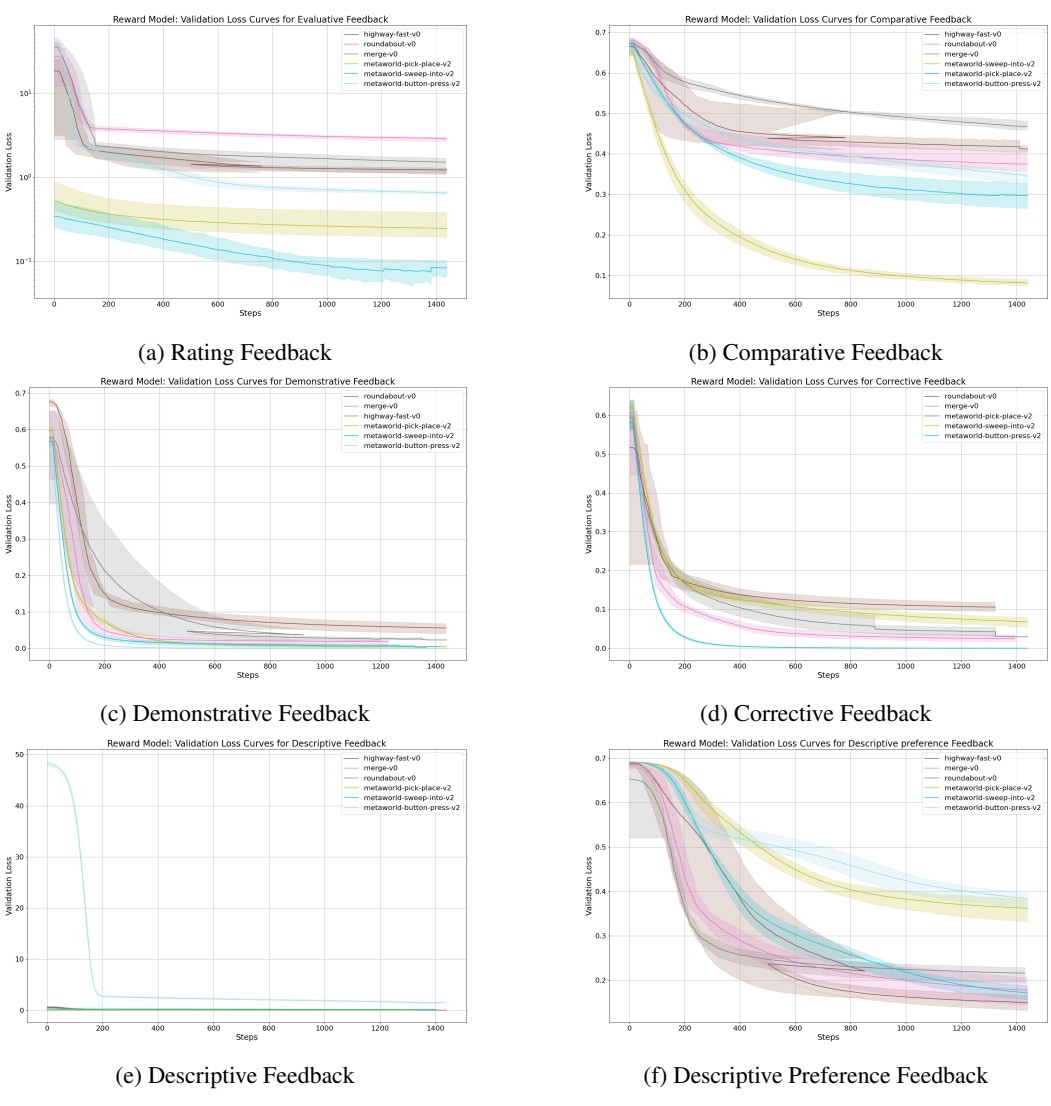

Figure 28: Continuation of Validation Loss Curves for Different Feedback Types (**Highway-Env**, **MetaWorld**): Steps is the number of optimizer steps. Averaged over five feedback datasets.

### D.4 REWARD MODEL TRAINING RESULTS WITH NOISE

As shown above, with optimal feedback, all reward models train effectively, achieving low validation loss. In the following plots, we show the reward model validation loss for different levels of noise, with $\beta = \{0., 1, 0.25, 0.5, 0.75\}$, averaged across 3 seeds. We can observe, that the introduced noise leads to a clear decrease in validation loss.

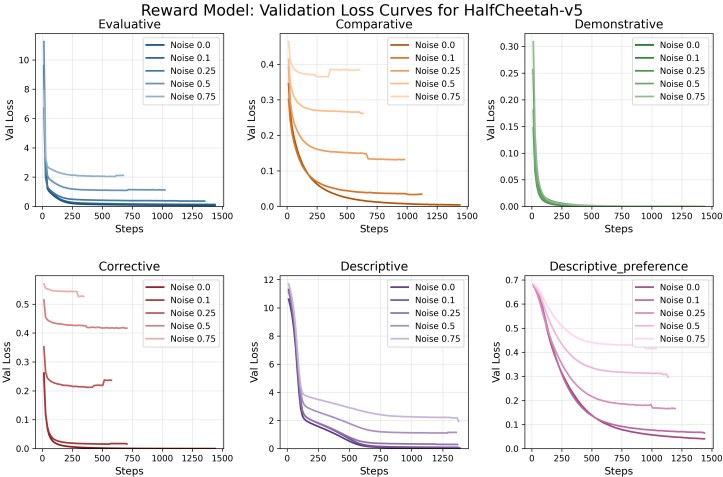

Figure 29: Reward Learning Loss Curve for **HalfCheetah-v5** at different levels of noise

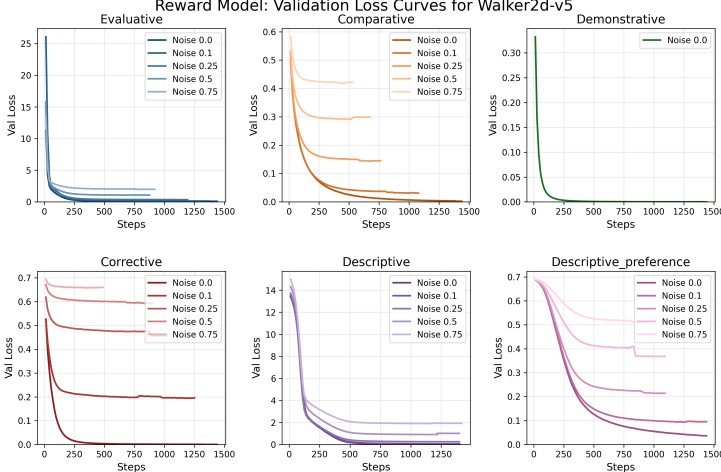

Figure 30: Reward Learning Loss Curve for **Walker2d-v5** at different levels of noise

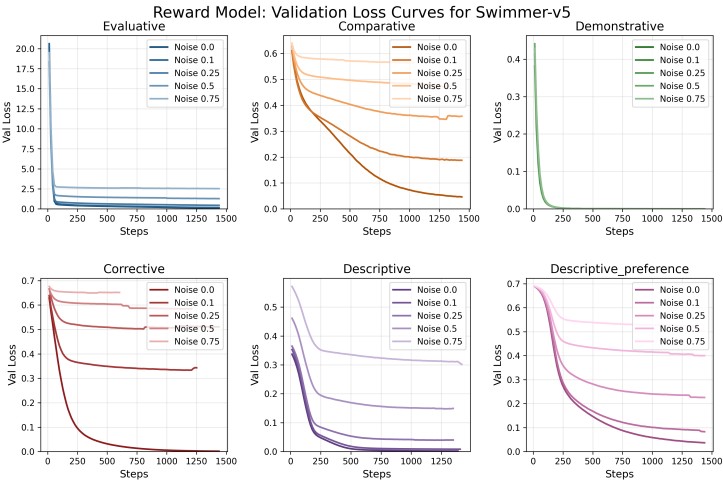

Figure 31: Reward Learning Loss Curve for **Swimmer-v5** at different levels of noise

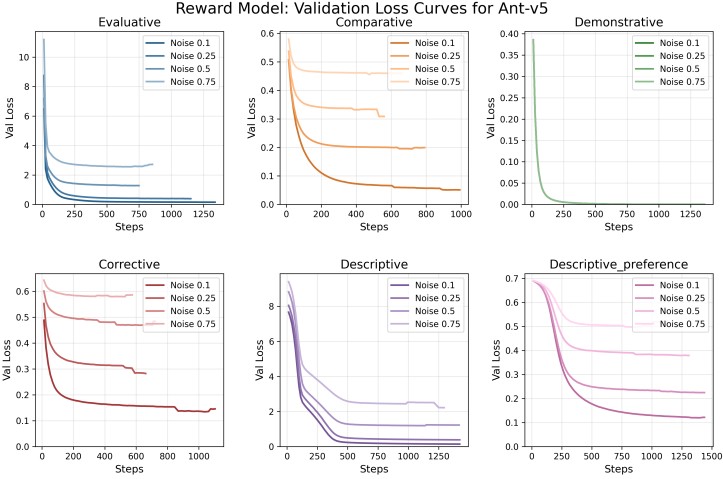

Figure 32: Reward Learning Loss Curve for **Ant-v5** at different levels of noise

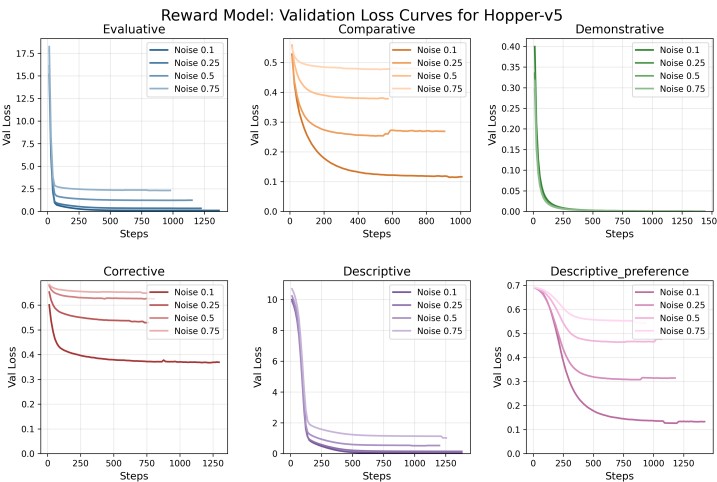

Figure 33: Reward Learning Loss Curve for **Hopper-v5** at different levels of noise

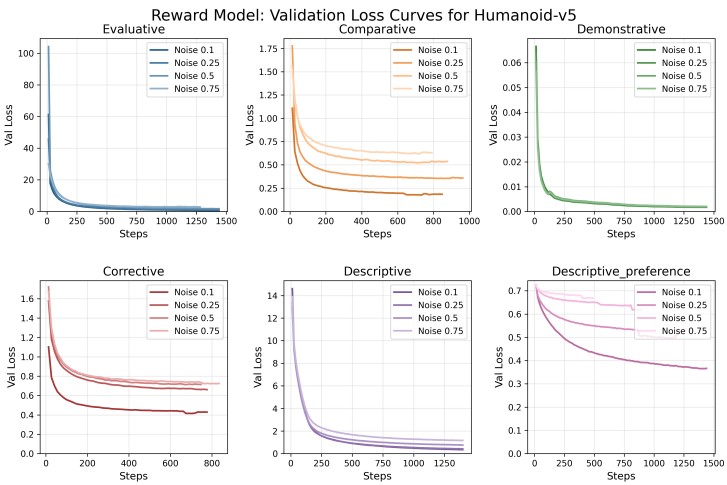

Figure 34: Reward Learning Loss Curve for **Humanoid-v5** at different levels of noise

## D.5 Correlation between Ground-Truth and Learned Reward Functions

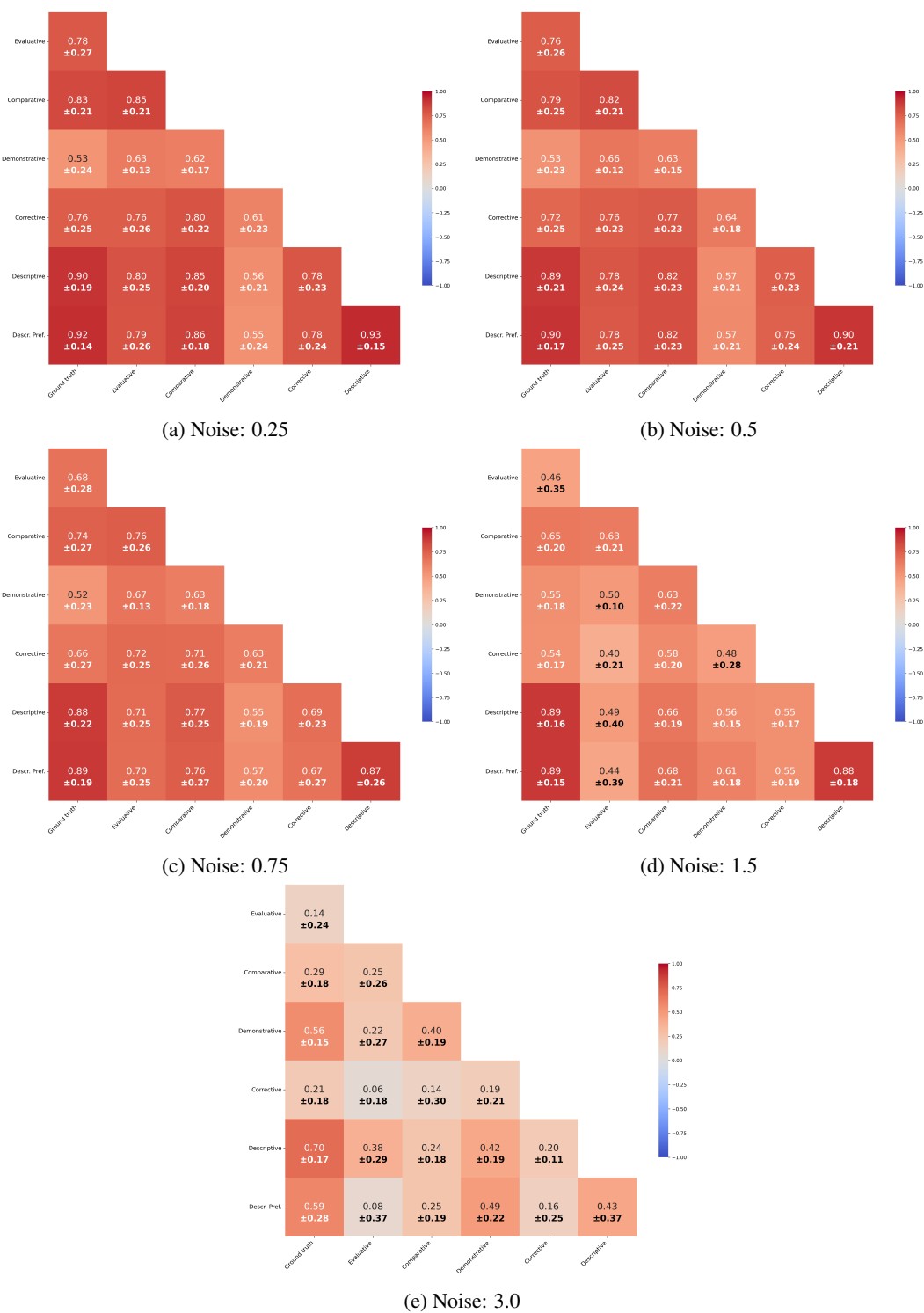

Figure 35: Summary of **pairwise reward function correlations** between feedback types at different noise levels ( Correlations are averaged across all six Mujoco environments, 3 seeds per env. and noise level)

### D.5.1 CORRELATION ACROSS FEEDBACK TYPES FOR INDIVIDUAL ENVIRONMENTS

(a) HalfCheetah-v5

(b) Walker-v5

(c) Swimmer-v5

(d) Ant-v5

(e) Hopper-v5

(f) Humanoid-v5

Figure 36: Pairwise correlations between the ground truth and learned reward functions from different feedback types. Optimal feedback without noise.

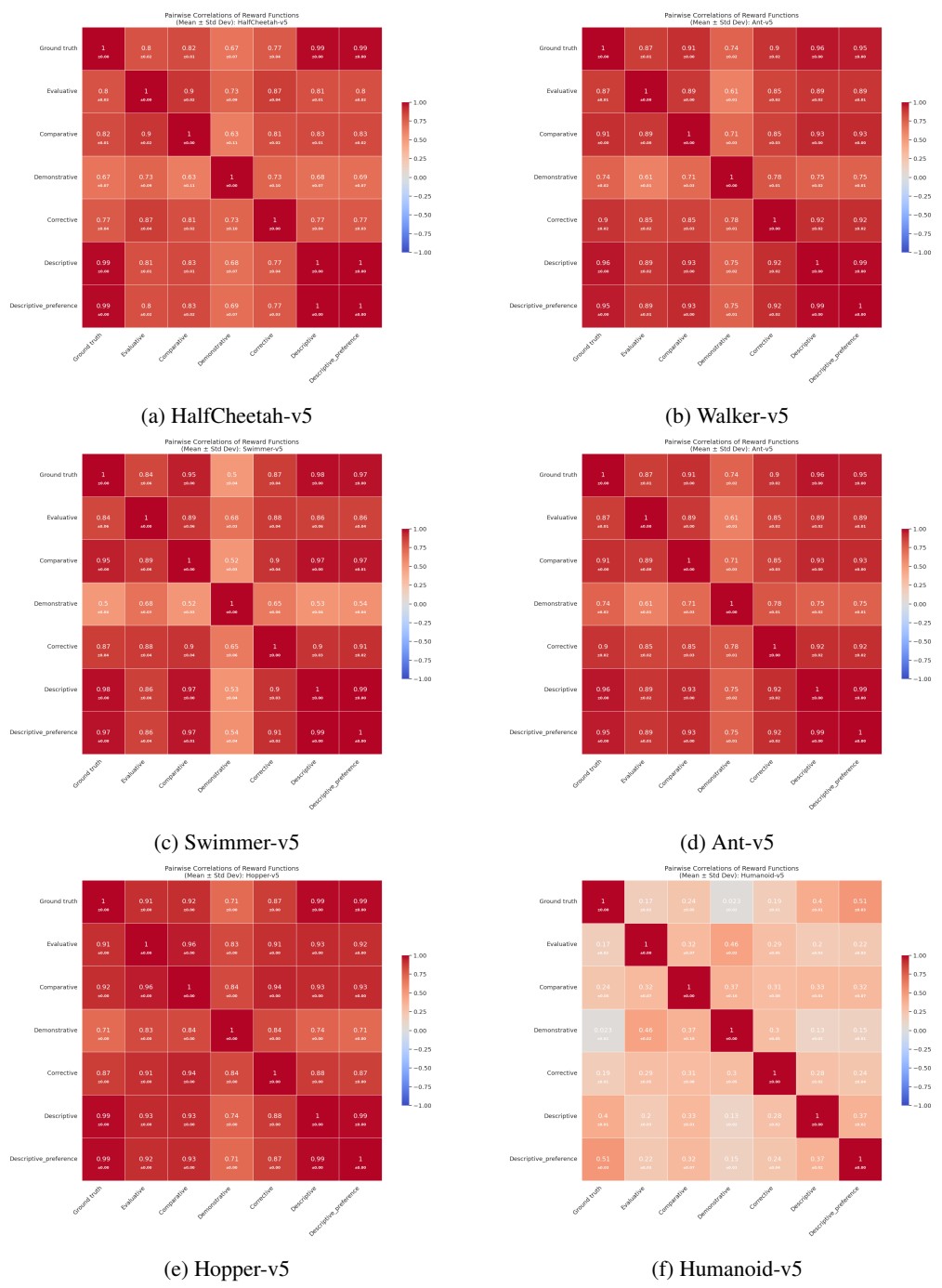

(a) HalfCheetah-v5

(b) Walker-v5

(c) Swimmer-v5

(d) Ant-v5

(e) Hopper-v5

(f) Humanoid-v5

Figure 37: Pairwise correlations between the ground truth and learned reward functions from different feedback types. Noise level 0.5.

## D.6 INFLUENCE OF NOISE ON CORRELATION WITH GROUND-TRUTH REWARD FUNCTION

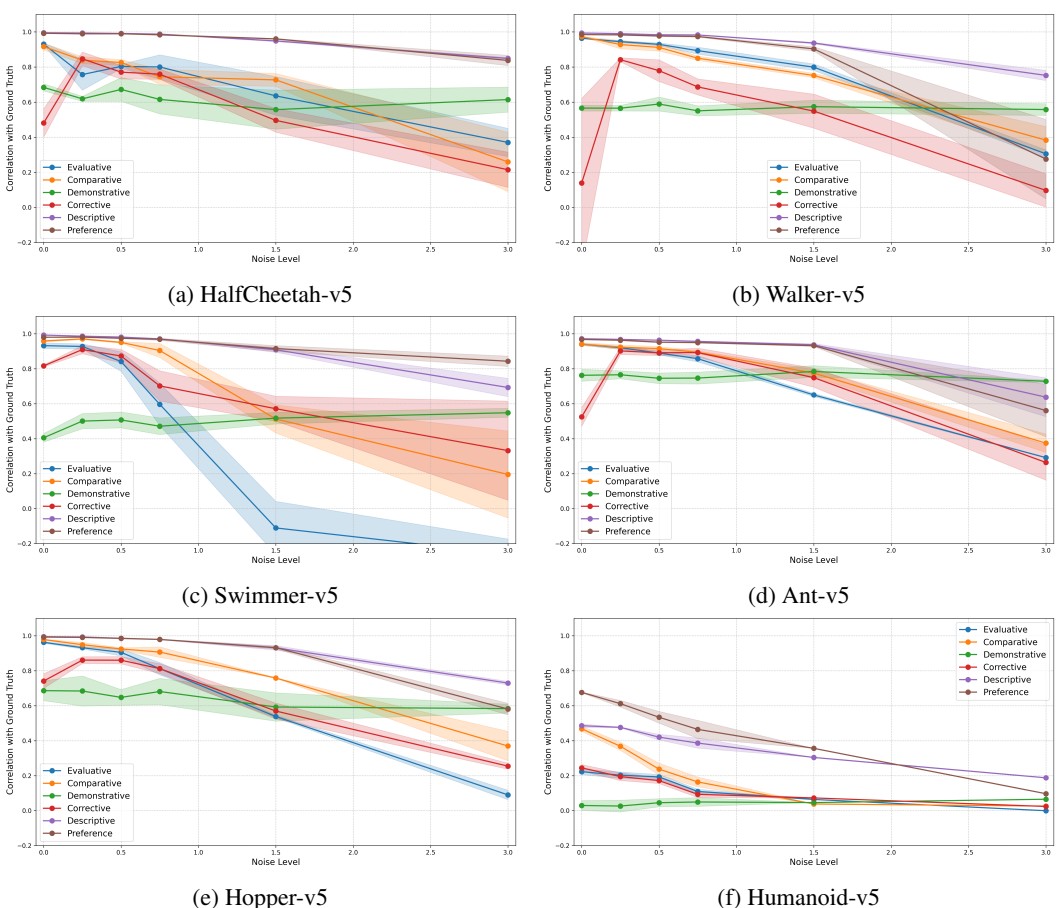

(a) HalfCheetah-v5      (b) Walker-v5

(c) Swimmer-v5      (d) Ant-v5

(e) Hopper-v5      (f) Humanoid-v5

Figure 38: Influence of noise on correlation with ground-truth reward function

## D.7 SEQUENTIAL REWARD PREDICTIONS

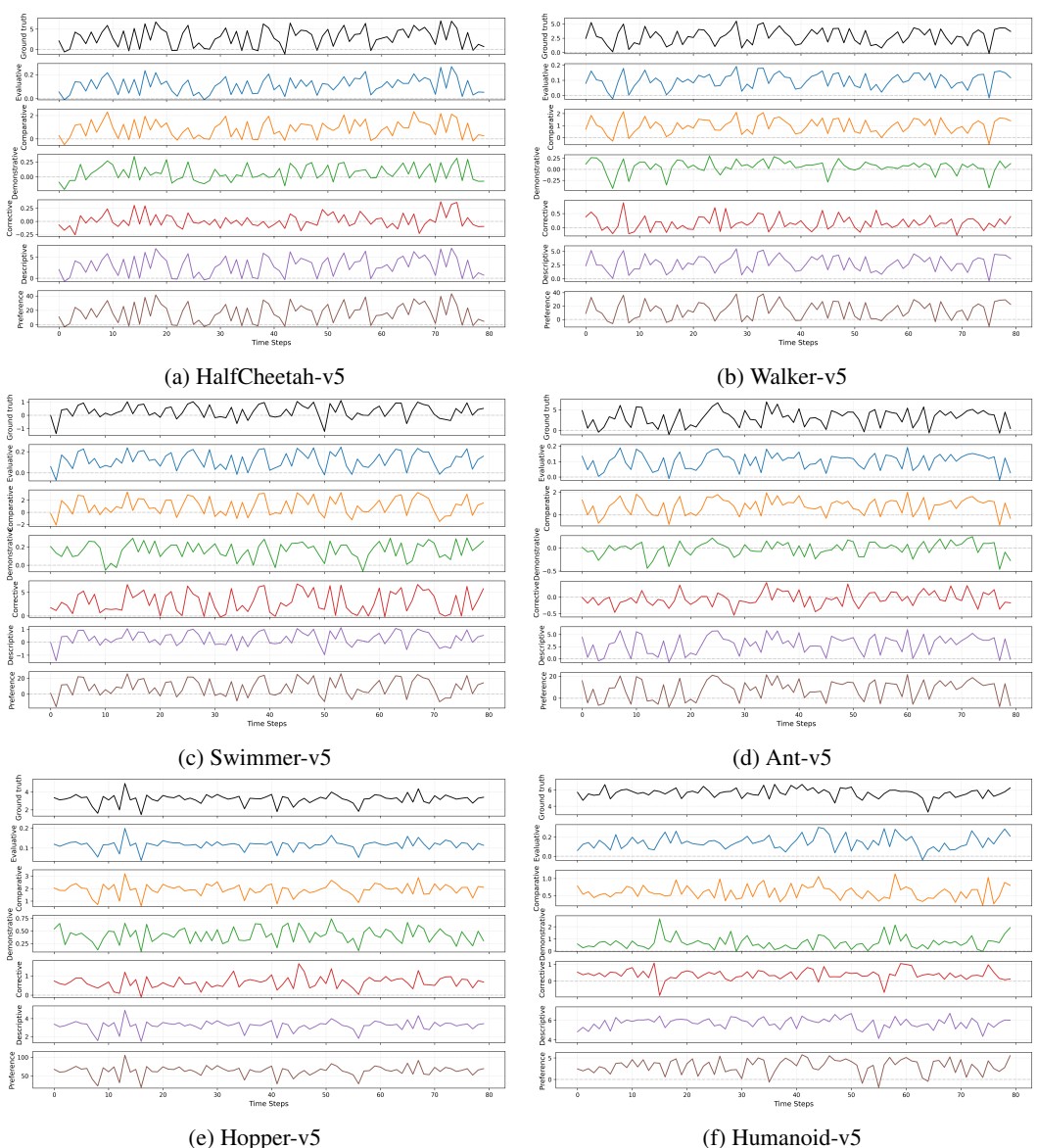

Figure 39: Ground-Truth reward function and reward model predictions for randomly sampled sequences.

# E  AGENT TRAINING DETAILS

## E.1  TRAINING CONFIGURATION

For the training of downstream RL-agents, we chose the same hyperparamter configuration as for the initial expert models (see Table 4 and Table 5).

We implemented the reward function as a wrapper around the *Gymnasium*-environment, which replaces the ground-truth environment rewards with the prediction of the reward model. A reward is predicted for each observed state-action pair. As mentioned above, for our analysis, we use a single pre-trained reward model. instead of a consciously updated reward model.
For the environments trained with PPO, we chose to standardize the rewards by subtracting a running mean, and dividing by the standard deviation. This follows the existing hyperparameters of RL for these environments, as available in *StableBaselines3-Zoo* (Raffin et al., 2021). The SAC agents are trained with unnormalized rewards for all runs.

## E.2  DETAILED REWARD CURVES FOR RL TRAINING WITH OPTIMAL REWARD MODELS

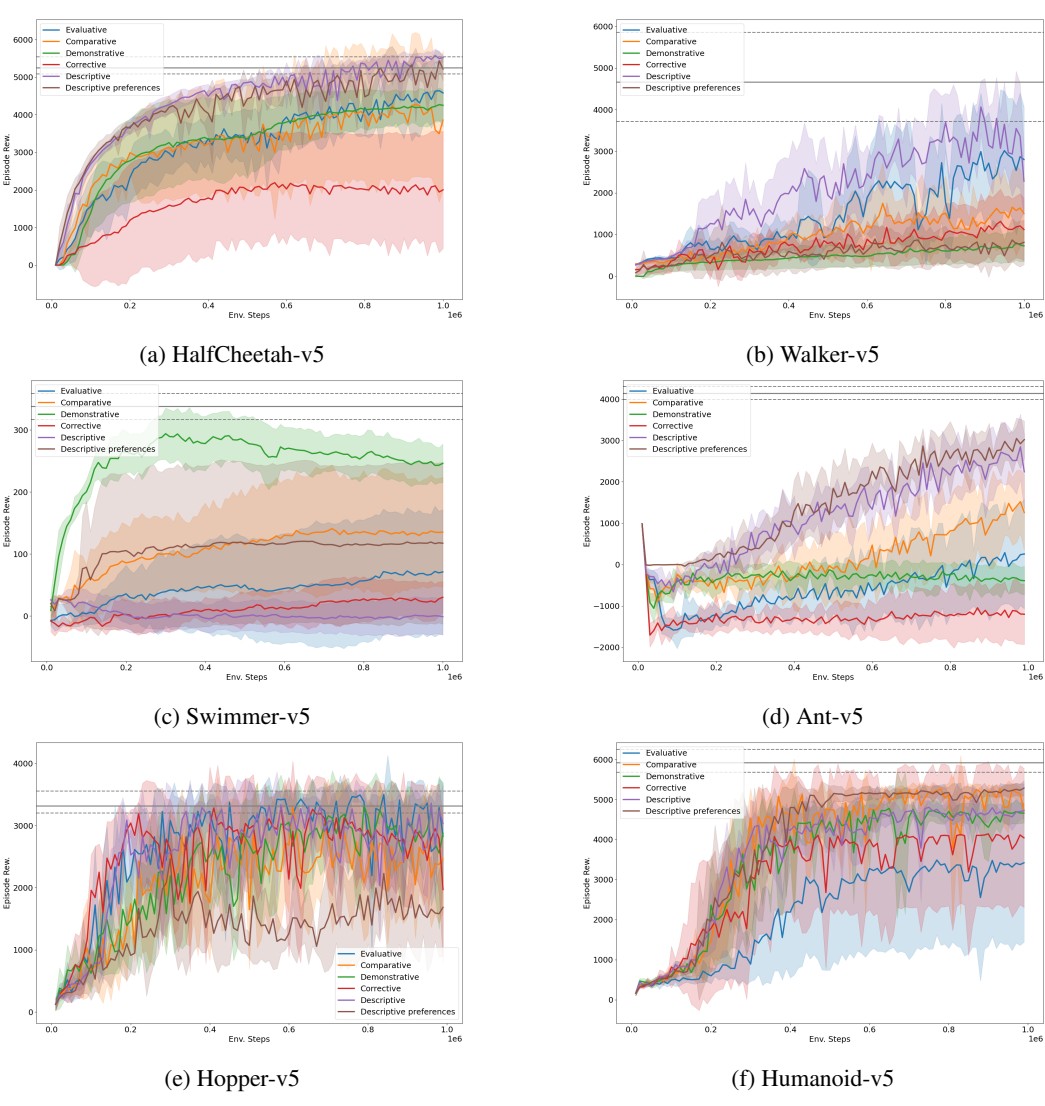

(a) HalfCheetah-v5

(b) Walker-v5

(c) Swimmer-v5

(d) Ant-v5

(e) Hopper-v5

(f) Humanoid-v5

Figure 40: RL reward learning curves for optimal rewards. The shaded area indicates the maximum and minimum reward curves. Rewards are aggregated over 5 feedback sets.

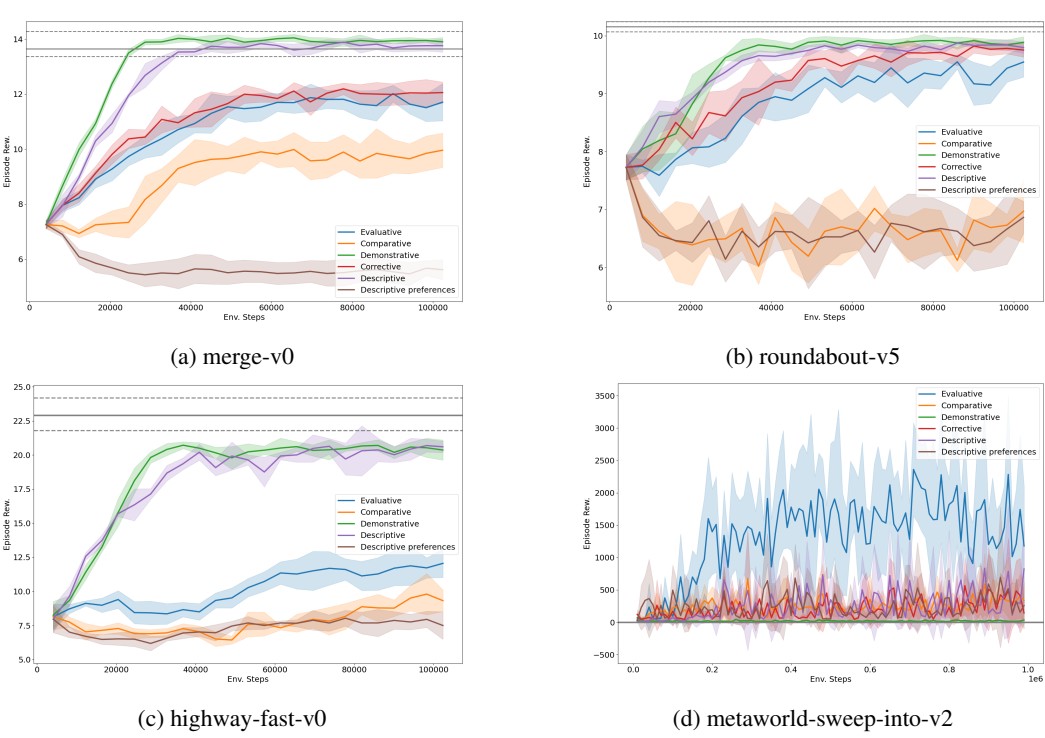

(a) merge-v0

(b) roundabout-v5

(c) highway-fast-v0

(d) metaworld-sweep-into-v2

Figure 41: RL reward learning curves for optimal rewards. The shaded area indicates the maximum and minimum reward curves. Rewards are aggregated over 5 feedback sets.

### E.3 DETAILED REWARD CURVES FOR RL TRAINING WITH NOISY REWARD MODELS

The following plots show the downstream RL performance for agents trained with reward function of multiple noise levels. Rewards are aggregated over 3 seeds per noise level. We can identify two trends: (1) Often, RL performance is surprisingly robust, even as reward model are way less accurate, and multiple feedback types are competitive in this regard. Descriptive feedback seems to be the most robust, only being heavily influenced by noise in one environment (*Swimmer-v5*), but actually improving its's performance over optimal feedback, which we attribute to an implicit regularization effect. (2) We do see that different environments can be more or less susceptible to introduced noise in general. Environments like **Hopper-v5** and **HalfCheetah-v5** seem more robust than an environment like **Swimmer-v5**.

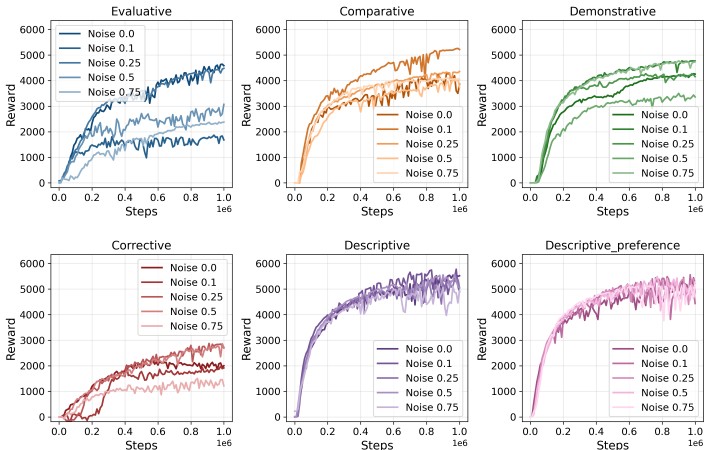

Figure 42: RL Reward Curves for **HalfCheetah-v5** at different levels of noise

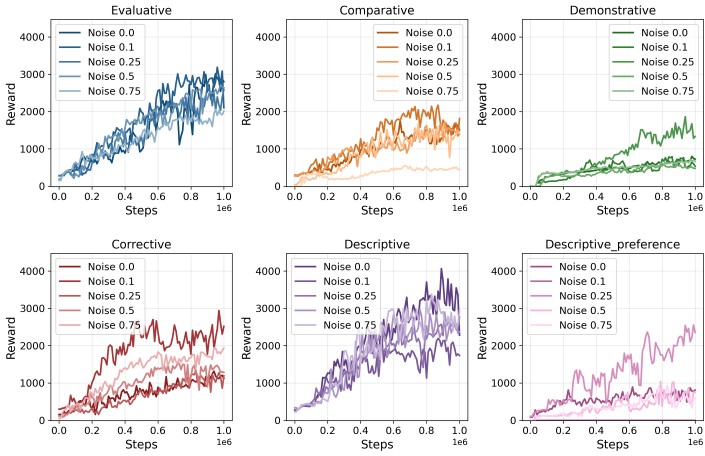

Figure 43: RL Reward Curves for **Walker2d-v5** at different levels of noise.

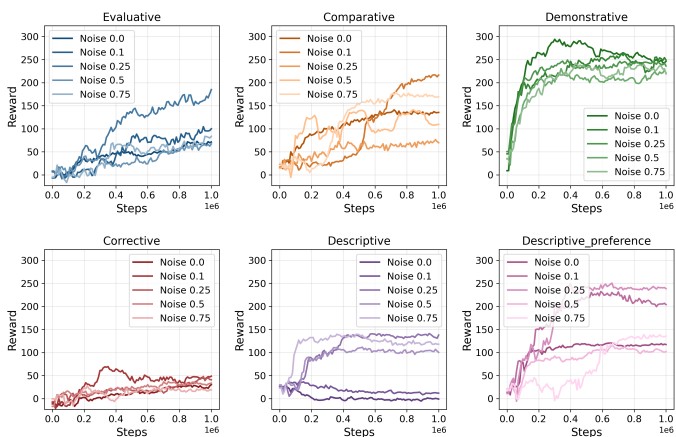

Figure 44: RL Reward Curves for **Swimmer-v5** at different levels of noise

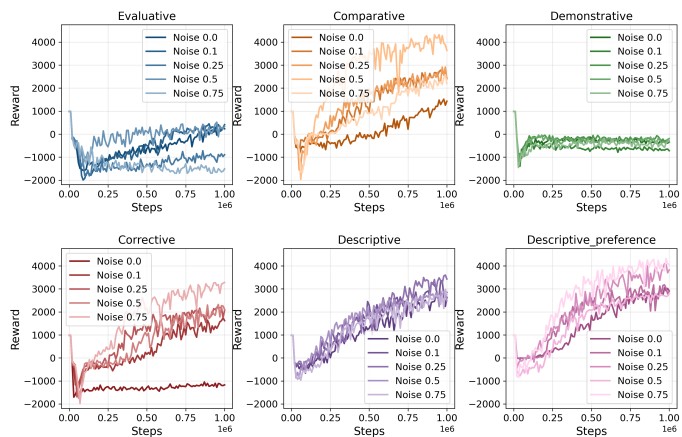

Figure 45: RL Reward Curves for **Ant-v5** at different levels of noise

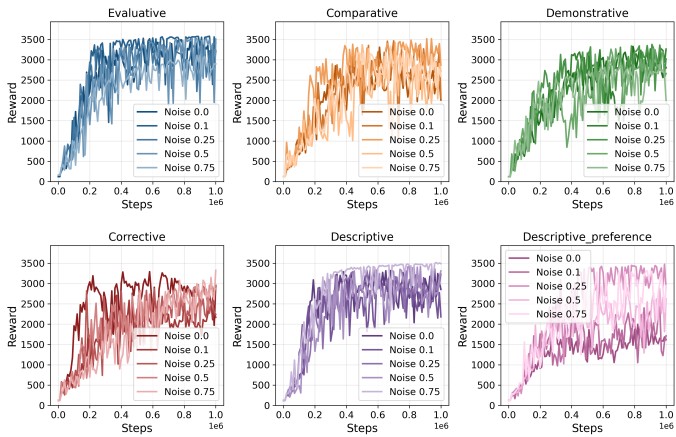

Figure 46: RL Reward Curves for **Hopper-v5** at different levels of noise

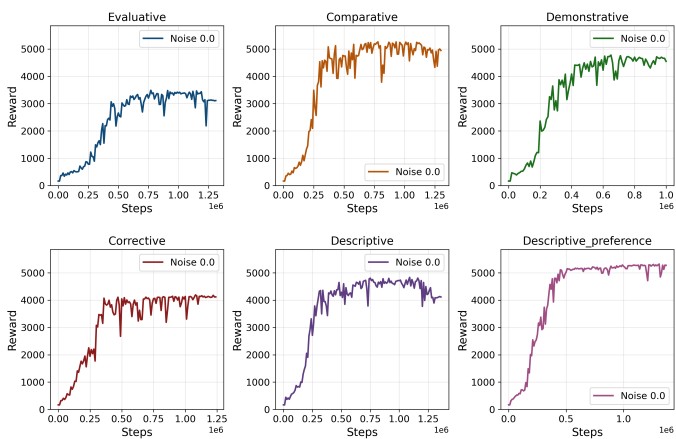

Figure 47: RL Reward Curves for **Humanoid** at different levels of noise

## E.4 Detailed Reward Curves for RL Training with Varying Amounts of Feedback

The following plots show the downstream RL performance for agents trained with varying amounts of feedback. We created datasets with fewer feedback instances by sampling random segments/state clusters from the full dataset. Rewards are aggregated over 3 seeds per amount of feedback.

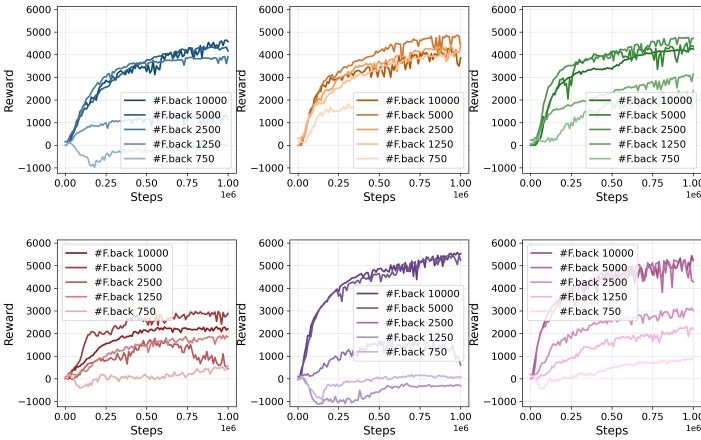

Figure 48: RL Rewards for **HalfCheetah-v5** with different amounts of feedback.

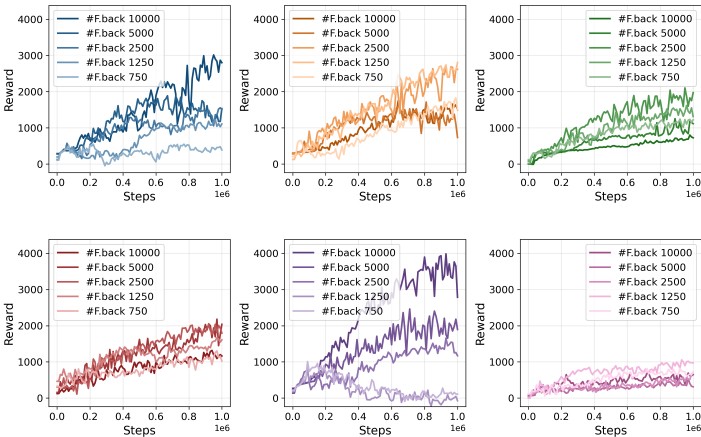

Figure 49: RL Rewards for **Walker2d-v5** with different amounts of feedback.

We find that all feedback types are sensitive to decreasing number of feedback instances. However we still can highlight some particular observations: Comparative feedback is relatively efficient, showing better performance with few feedback instances. Other types, like evaluative and descriptive feedback show a noticeable drop-off for fewer instances. However, sub-sampling might impact evaluative and descriptive feedback types in an particular manner: Because the reward histograms, and the clustering are computed for the entire dataset, sampling from the full dataset is not representative of recomputed statistics/a recomputed coarse clustering. We therefore report this experiment as an important indictor for the stability of different feedback types, and encourage future exploration.

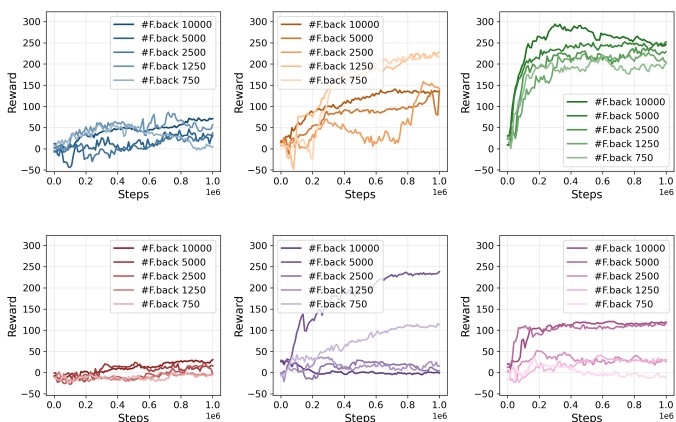

Figure 50: RL Rewards for **Swimmer-v5** with different amounts of feedback.

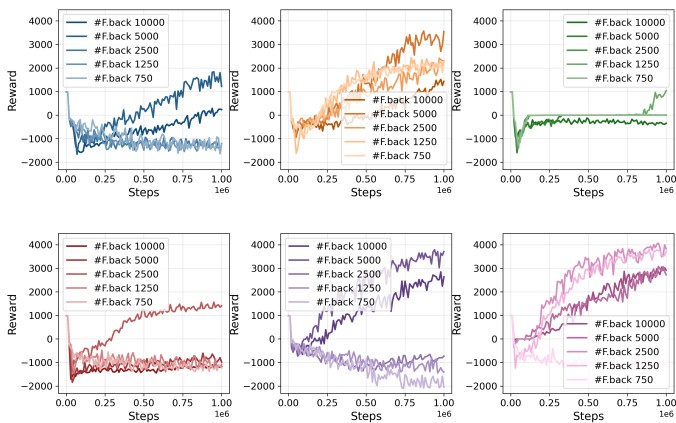

Figure 51: RL Rewards for **Ant-v5** with different amounts of feedback.

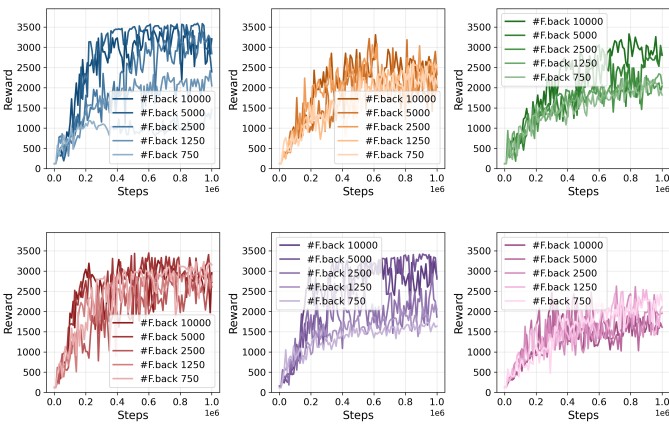

Figure 52: RL Rewards for **Hopper-v5** with different amounts of feedback.

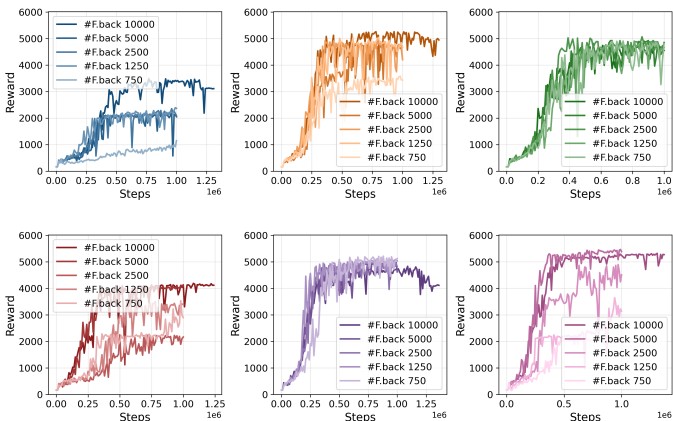

Figure 53: RL Rewards for **Humanoid** with different amounts of feedback.

### E.5 BEHAVIORAL CLONING BASELINES

For each environment, we also train a baseline via supervised behavioral cloning (BC). As the training dataset, we utilize the demonstrations, generated for the feedback data. Segments are flattened, i.e. 10,100 segments of 50 steps each lead to 500,000 state-action pairs for training of the policy. For the policy network, we use a two-layer neural network with 32 units (as is the default in the *imitation* library Gleave et al. (2022)). Additionally, entropy-regularization is used to avoid over-fitting of the supervised policies. We have trained networks with a batch size of 32, and 20 maximum epochs on the dataset. However, we often did not see significant improvement after around 5 episodes.

## F   JOINT-REWARD MODEL DETAILS

### F.1   TRAINING CONFIGURATION

Similarly to the other training runs, we chose default training parameters. However, more extensive hyper-parameter search might yield better performance in the future.

For training, the RL algorithm is unmodified from a normal training setup. We only adapted the reward function. As before, we implement learning with the multi-type ensemble via a wrapper around the environment. This time, instead of a single reward function, we load a set of reward functions.

For a state-action pair, each individual reward model is queried and predicts a reward. As mentioned above, we standardize the individual reward functions, as they can be in dissimilar value ranges. We do this by standardizing each individual reward function, by subtracting the rolling mean, and dividing by the rolling standard deviation. We compute the rolling standard deviation with Welford's algorithm Welford (1962).

Given the ensemble of single feedback reward functions $R_{fb} = \{\hat{r}_{eval}, \hat{r}_{comp}, \hat{r}_{demo}, \hat{r}_{corr}, \hat{r}_{descr}, \hat{r}_{descr.pref.}\}$ we implement two variants:

- Averaging: We just average the normalized reward into a joint reward prediction

$$r(\hat{s}, a) = \frac{1}{|R_{fb}|} \sum_{r_{type} \in R_{fb}} r_{type}(s, a)$$

- Uncertainty-Weighted Ensemble: Each single feedback reward model is an ensemble in itself (i.e., relying on the Masksembles architecture). We can therefore receive an approximate posterior probability for each feedback type, i.e. a mean and standard deviation between the sub-model predictions. We experimented with using these standard deviations as uncertainties for each feedback type. We then weighted the reward predictions with the inverted standard deviation, i.e., a feedback type with lower uncertainty for a state-action

pair contributes to the prediction.

$$\hat{\mu} = \frac{\sum_{i=1}^{n} \frac{\mu_i}{\sqrt{\sigma_i^2}}}{\sum_{i=1}^{n} \frac{1}{\sqrt{\sigma_i^2}}}$$

