# OpenReview forum: "Reward Learning from Multiple Feedback Types"
_ICLR.cc/2025/Conference — ICLR 2025 Poster_

### Official Review · Reviewer_vPP3 · 2024-10-29

**Soundness:** 2
**Presentation:** 1
**Contribution:** 2
**Rating:** 8
**Confidence:** 4

**Summary:**

This paper explores reward learning utilising multiple different types of feedback, covering evaluative, instructional, and descriptive forms.

They propose a method on how to generate these forms of feedback synthetically, how to learn from these feedback methods in isolation, and also how to combine these reward signals to learn from many types of feedback simultaneously.

Additionally, they investigate the different properties of these feedback methods such as their correlation, and ability to provide a reward signal for learning an RL policy.

**Strengths:**

Research is well motivated.

Existing work is nicely leveraged and inter-twinned to form basis of paper without re-inventing the wheel.

Paper is reasonably well written, with ideas are clearly communicated.

The proposed method to generate synthetic feedback seems good.

The proposed method to learn from multiple feedback types, namely training separate reward models and combining their scores, is clean, simple, and easy to understand.

The noise model used seems reasonable.

Paper considers alternative formulations of human feedback, namely regret-based.

Exploring the reward model correlations is good analysis.

For the most part, the paper tests on many environments with many noise levels, giving strong evidence for some of their claims.

**Weaknesses:**

Learning a reward model for each feedback type presents some limitations not considered by the authors.
* If these reward models are large or computationally expensive (e.g. in an foundation model finetuning setting where each one may be foundation model sized), then having one for every feedback type might not be very scalable or practical.
* The resulting reward signal cannot incorporate information only deducible from considering multiple feedback types at once.
For example, there may be some patterns or generalisations only apparent when considering all feedback.
* If there's only small amounts of one type of feedback, its corresponding reward model might be very inaccurate, and if the uncertainty for the uncertainty-weighting is not well calibrated, it could lead to that reward model making the overall reward signal worse.

The paper only considers synthetically generated feedback and does not evaluate / verify with real human feedback.

The paper focuses on continuous control tasks, and does not test on any discrete or contextual bandit environments.
This is important to test as this more closely corresponds to the LLM fine-tuning setting, a key application of RLHF and related methods.

Interpreting the demonstrations as being preferred only to random trajectories seriously limits the information extracted from them compared to standard demonstration learning approaches like MaxEnt.
Despite the authors claims suggesting this was "stronger than sampling against other sub-optimal rollouts", the amount of interesting state space explored by random policies decreases exponentially as the environment gets larger and more complicated.
For example, in the language modelling setting, interpreting your demonstrated responses as better than random tokens would probably not improve your pre-trained model compared to supervised fine-tuning.

The feedback methods, except one, all use the same fixed segment length.
This may be sub-optimal if different feedback methods work better with different sized segments.

The reward model correlation analysis is only done on one environment and are averaged over the whole trajectory.
It's not clear how consistent these correlations are across different environments, and it may be interesting to see how these correlations change over the course of a typical trajectory (E.g. maybe some are well correlated to begin with, and then become de-correlated).

Graphs have no error bars, and it's not clear how many random seeds have been averaged over in the RL experiments.
If there are not multiple random seeds, then very little of the results presented can be assumed to be statistically significant.

The reward models are pre-trained before being used to train a policy.
This is very atypical for RLHF and limits the conclusions that can be drawn from the results.

In figure 3, not showing the score for learning from the ground truth reward as a function of environment time steps limits comparisons between learning from the feedback modes vs the ground truth itself.

It's not clear how amounts of each type of feedback are controlled to be comparable.
It seems the amount of each type available is very dependent and sensitive to parameters of the synthetic generation process.

For uncertainty-weighted ensembles, the uncertainty of an ensemble that has been trained on feedback that only constrains reward differences (e.g. comparative) is not well-calibrated.

I do not believe some of the claims in the discussion and conclusion are well-supported.
* On point (2), line 495/496, combining rewards is only tested on two environments and only performs well in one of them.
* Line 523/524, the comparison of these characteristics has been very limited.
* Line 526 to 528, learning from multiple feedback types being "very effective" is not supported by the evidence presented in figure 6.

## Errata:
* Figure 2 caption and subcaptions disagree on what environment the results are from (Swimmer-v5 vs HalfCheetah(-v5)).
* Figure 3 partially covers some text
* The paragraph at the start of section 4.4 conflicts with itself: "instead of continuously adapting ... is continuously updated"
* Line 380, sentence abruptly ends and is unfinished
* The axes and legends for figures 2,3,4, and 5 (especially 3), are too small to clearly read, especially on a printed copy of the paper.
* The lighter shaded lines in figures 4, 5, and 6 are hard to see.
* There is no reference to figure 4 in the main text of the paper, nor analysis of what it shows.
* Line 454, "As stated before, each individual reward model ... is an ensemble in itself". This does not appear to have been stated before.
* Line 467, the text refers to the "Hopper-v5" environment, but the figure under discussion, 6, only contains the HalfCheetahv5 and Walker2d-v5 environments.

**Questions:**

Does the method of reward modelling proposed work to learn both an RL policy and a reward model *without pre-training the reward model*?

When utilising demonstrative and preference feedback, a common method is training first on the demonstrations, e.g. using MaxEnt or SFT / Behavioural cloning, and then fine-tuning on the preferences with RLHF.
It would be interesting to see a comparison to this baseline method.

How do reward function correlations vary across environments and across trajectories?

It would be good to run multiple seeds and plot mean and standard error/deviation in the figures.

---

> ### Author Response · Authors · 2024-11-24
>
> We thank the reviewer for their comments and helpful suggestions. We have summarized the steps for a revised version in a summary comment. We still want to answer the points raised in this review briefly:
>
> "If these reward models are large or computationally expensive."
> -> We see our ensemble approach as a proof of concept at this stage; in the future, using multiple heads on top of a shared model (e.g., compared to bootstrapped DQN) might be a practicable solution, and we want to encourage future exploration strongly, we raise the issue of expanding our approach for LLMs
>
> "The resulting reward signal cannot incorporate information that is only deducible from considering multiple feedback types at once. For example, some patterns or generalizations may only be apparent when considering all feedback."
> -> Can the reviewer kindly clarify this point? As multiple reward functions are queried during inference (which might have been trained on the same instances with different feedback), we see the possibility of incorporating different information
>
> "If there are only small amounts of one type of feedback, its corresponding reward model might be very inaccurate, and if the uncertainty for the uncertainty-weighting is not well calibrated."
> -> This is a very important issue that we want to address with our uncertainty-weighted approach, and we would acknowledge that the models at very early training stages might be very inaccurate due to random initialization; possible strategies might be the use of pre-training or initialization techniques
>
> The paper only considers synthetically generated feedback and does not evaluate/verify with real human feedback.
> -> We acknowledge this as a weakness of the paper. An extension of our approach by fitting the characteristics/noise levels with human data is an intriguing future expansion of our work, and we have noted this in future work. However, we would argue that our study still provides novel insights beyond the toolkit, as the utility of multi-type feedback has not been shown before on a conceptual or empirical level.
>
> "The paper focuses on continuous control tasks and does not test on discrete or contextual bandit environments."
> -> We have added Highway-Env as a discrete RL environment and found results to be consistent with previous results
>
> "For example, in the language modeling setting, interpreting your demonstrated responses better than random tokens would probably not improve your pre-trained model compared to supervised fine-tuning."
> -> The reviewer is absolutely correct that this pattern does not need to hold across more complex environments, although we find it very reliable in our experiments. This kind of adaptation is possible with our framework, and we would like to encourage future work on this.

---

> > ### Author Response · Authors · 2024-11-24
> >
> > "The feedback methods, except one, all use the same fixed segment length. This may be sub-optimal if different feedback methods work better with different-sized segments."
> > -> We will add this to the limitations and suggestions for future work and choose this segment length mainly based on the justification given in previous work that longer segments are easier to label for humans
> >
> > "The reward model correlation analysis is only done on one environment and is averaged over the whole trajectory."
> > -> We thank the reviewer for this suggestion; in response, we have added more details and a temporal analysis of reward functions
> >
> > "If there are not multiple random seeds, then very little of the results presented can be assumed to be statistically significant."
> > -> We acknowledge that we should have provided these in the initial version of the main paper (as all experiments were run over multiple seeds). We have just provided them in the appendix. We have added error bars/confidence intervals back to the reworked figures in the main paper.
> >
> > "In Figure 3, not showing the score for learning from the ground truth reward as a function of environment time steps limits comparisons between learning from the feedback modes vs. the ground truth itself."
> >
> > -> Again, this was an oversight on our part. We indicate the expert performance based on the ground-truth reward function, but it was difficult to see within the figures. We have updated the figures to improve visibility.
> >
> > "The reward models are pre-trained before being used to train a policy. This is very atypical for RLHF and limits the conclusions that can be drawn from the results."
> > -> While not typical, this approach has been used in previous work and simplified the analysis space. However, we acknowledge this as a weakness and plan to expand this in future work.
> >
> > "It is not clear how amounts of each type of feedback are controlled to be comparable. It seems the amount of each type available is very dependent and sensitive to parameters of the synthetic generation process."
> > -> Can the reviewer kindly clarify this point? We try to exactly match the number of queries/feedback instances for each type of feedback (i.e., 10.000 preference pairs, demonstration segments, ratings, etc. for our experiments) and average our results over five datasets to control for the composition of feedback datasets
> >
> > "For uncertainty-weighted ensembles, the uncertainty of an ensemble trained on feedback that only constrains reward differences (e.g., comparative) is not well-calibrated."
> > -> We will note this as a limitation
> >
> > "I do not believe some of the claims in the discussion and conclusion are well-supported."
> > -> We tried to sharpen our claims based on the additional results in section 4
> >
> > Errata
> > -> We thank the reviewer for pointing out these errors. We have tried to address them for a revision carefully

---

> > > ### Comment · Reviewer_vPP3 · 2024-11-25
> > >
> > > Thank you for the clarification and the updated paper. The changes made largely address my concerns and improve the strength of the paper, thus I have altered my rating.
> > >
> > > To clarify some things from my review:
> > >
> > > ""The resulting reward signal cannot incorporate information that is only deducible from considering multiple feedback types at once. For example, some patterns or generalizations may only be apparent when considering all feedback." -> Can the reviewer kindly clarify this point?"
> > > What was meant here is that there may be some aspect of reward that might only be deducible by a single reward model learning from multiple feedback types simultaneously. As a slightly contrived toy example, consider a 2D real-valued state space with two reward functions, r1 and r2, both trained on a different preference. r1 sees ([1, 0] > [-1, 0]) and r2 sees ([0, 1] > [0, -1]). r1 implements r1([x, y]) = 1 if x > 0 else 0, and r2 implements r2([x, y]) = 1 if y > 0 else 0. Now consider r3 which sees both preferences and learns r3([x,y]) = 1 if x+y > 0 else 0. (r1+r2)/2 != r3. Whilst the learnt rewards share similarities, they are somewhat different due to generalising differently. This is only a minor point, but one I thought worth pointing out. Note the different preferences are stand-ins for different types of reward, I'm aware in practice for this example a single preference model would see both the preferences.
> > >
> > > ""It is not clear how amounts of each type of feedback are controlled to be comparable. It seems the amount of each type available is very dependent and sensitive to parameters of the synthetic generation process." -> Can the reviewer kindly clarify this point?"
> > > Your explanation clarifies what I was trying to get at, thank you.

---

> > > > ### Comment · Reviewer_vPP3 · 2024-11-30
> > > >
> > > > Thank you for the additional figures and appendices, they are quite interesting.
> > > >
> > > > Based on this, and additional changes made, I have raised my score.

---

### Official Review · Reviewer_oTK8 · 2024-10-31

**Soundness:** 2
**Presentation:** 2
**Contribution:** 3
**Rating:** 3
**Confidence:** 5

**Summary:**

The paper presents a benchmark or toolkit to use to simulate multiple types of feedback in the context of learning preferences from human feedback as well as analyzes one approach to combining multiple sources of feedback (ensemble of rewards). The different sources of feedback include ratings (scalar scores per trajectory), binary preference labels, demonstrations, corrections, descriptions of high value state-action pairs, and descriptions of high value features. The paper describes in detail how each sources of feedback is simulated using an environment's ground truth reward as the human proxy. The bulk of experiments focus on comparing the reward models learned from the different sources of feedback by looking at correlation between learned reward functions and downstream policy performance according to the ground truth reward. Policy performance and reward model accuracy are evaluated in the face of feedback noise, with how noise is applied varying between the different sources of feedback. For evaluating how to combine the multiple sources of feedback, the authors present two approaches to combining an ensemble of reward models, where different mini reward model ensembles are trained from different sources of feedback. The primary take away is that there is no one best source of feedback across tasks.

**Strengths:**

- The paper addresses an important problem, which is providing a toolkit/benchmark for people to use to research learning from different types of feedback and how to combine them. The approach is similar to what has already been used to learn from binary preference labels alone, which makes it an easy toolkit for people to pick up and understand how it works.
- The experiments and results demonstrate that accounting for multiple source of feedback is neither straightforward nor trivial, and work is needed to understand the strengths and benefits of each.
-The proposed toolkit does not rely on actual humans in the loop making it something many people can use easily for initial proofs of concepts.
- The toolkit provides mechanisms to have different sources of noise, which is crucial as we know human feedback is noisy. The code is note yet provided, but from the description in the paper, it seems like running experiments with different amounts of noise should be fairly straight forward.

**Weaknesses:**

**High-level overview:**

There are two main weaknesses of this paper discussed at a high level here, but more details below. The work is valuable and necessary, but the needed level of rigor isn't there yet.
     (1) the toolkit it not validated against any studies involving humans, therefore it is impossible to know if the conclusions drawn in the paper reflect characteristics of the feedback or of the implementation
     (2) the text in the paper, especially in the second half describing experiments and results, contradicts itself and presented results making it difficult to draw conclusions and have take aways or learnings.

**The details:**
* The way the paper is written suggests that both the multi-feedback toolkit and the method for combing the multiple sources of feedback are both equal contributions. I would push back that the main contribution is the toolkit and the proposed method is rather trivial with no true baselines to show its benefit or contribution. It would have been better to limit the feedback combination methods as a tool for proof of concept showing how the toolkit can be used.
* The authors draw conclusions about the usefulness of different types of feedback (e.g. Figure 2 reward model correlation and demonstrative and corrective having the worst correlation with the ground truth reward). However, in the absence of a study to validate the toolkit against human feedback makes it impossible to validate if this is a function of the feedback, or the choices they have made in how the feedback is implemented.
* Only one environment is evaluated therefore it is not possible to assess how general the proposed toolkit and feedback simulation method are. At least one other should be included. MetaWorld is popular for the preference learning and would be a natural fit as it is included in [BPref] (https://github.com/rll-research/BPref).
* There are multiple places in the paper where the authors seem to contradict either themselves or the presented results:
     * at the start of Section 4.4, the first sentence sounds like there was not online reward model adaptation, but then the last two sentences to that first paragraph make it sound like there was: "...we utilize simple pre-trained reward models instead of continuously adapting the models..." versus "The reward model is continuously updated..."
     * the text in the main body does not mentioned that the reward model correlations for other tasks beyond half cheetah have are weaker with different patterns among the feedback types. The full nuance of the results are not represented, and instead strong language and conclusions are drawn about a single task.
     * the statement on lines 325 - 326 "...all feedback types individually can learn reward functions that closely match the ground-truth reward..." is not true for all feedback types according to the Figure 2 and those in the appendix. In Figure 2, demonstrative and corrective have a correlation of 0.61 and 0.5, which is a medium match. In Figure 25, these feedback types have correlations as low as a 0.18.
     * on lines 410 - 411, it is stated that "...no feedback type is obviously more or less robust to noise." however, then looking at Figure 4(b) and Figure 5(b), the performance gap between learning curves across different amounts of noise varies across feedback types, which suggests that some are more sensitive that. others. To back up the claim, "obviously" needs to be quantified to make it clear the threshold for what qualifies as "more or less robust".
     * section 5 opens by stating that different feedback types struggle in different scenarios, but all of the discussion in Section 4 talks about how there is no clear difference between the different sources of feedback.
* The first half of the paper motivating and describing the multiple feedback toolkit is very well written, methodical, and easy to follow. However, there is a switch part way through the paper when it transitions to talking about experiments with the toolkit. Here the presentation and writing changes drastically with things like:
     * Figure 3 overlapping the text of the main body
     * an incomplete sentence (page 8 line 380)
     * what seems to be misconnected analysis and results figures (e.g. figures 4 and 5 - the text seems to talk about figure 4, but references figure 5 and figure 4 is not mentioned in the paper as far as I can tell)
     * mislabelled and titled figures where the captions disagree with the figure axis/title (e.g. Figure 2 - HalfCheetah-v5 versus Swimmer-v5; Figure 5 - swimmer v5 and walker2d-v5 versus half cheetah-v5; and figure 4 - "reward model validation curves", but it looks like policy returns)
     * Figure 4 left plot, the y-axis scale are on different scales, making it tricky to compare across feedback sources.
     * using both a : and - on page 8 line 429
     * it is stated that results are over 5 random seeds, but there are no standard deviation results
     * there are numerous places with typos (e.g. "as the be considered" line 256) that need to be addressed.
     * not all lines in Figure 3 are labelled nor described.

**Questions:**

* In Figure 5, the results are described as averaged over "3 feedback datasets", what are the sources of the different feedback datasets? Are these different random seeds? Earlier it was stated that 5 random seeds were used.
* How well do the feedback ensemble rewards correlate with the ground truth reward function?
* In Section 4.2 you talk about sampling against random behavior being stronger than agains sub-optimal rollouts for demonstrative feedback. This is an interested conclusion and I would have expected the opposite. Why do you think this is the case?

---

> ### Author Response · Authors · 2024-11-24
>
> We thank the reviewer for their comments and helpful suggestions. We have summarized the steps for a revised version in a summary comment. We still want to answer the points raised in this review briefly:
>
> “It would have been better to limit the feedback combination methods as a tool for proof of concept showing how the toolkit can be used.”
> -> We acknowledge this as a very fair comment and agree with this statement; we have thus adapted our stated contribution to mark it as a proof of concept and have, in turn extended the analysis of reward functions in section 4
>
> “the absence of a study to validate the toolkit against human feedback makes it impossible to validate if this is a function of the feedback or the choices they have made in how the feedback is implemented.”
> -> An extension of our approach by fitting the characteristics/noise levels with human data is an intriguing future expansion of our work, and we have noted this in future work. However, we would argue that our study still provides novel insights beyond the toolkit, as the utility of multi-type feedback has not been shown before on a conceptual or empirical level.
>
> “Only one environment is evaluated; therefore, it is impossible to assess how general the proposed toolkit and feedback simulation method is.”
> -> We have added additional results for Highway-Env and Metaworld
>
> “The first sentence sounds like there was no online reward model adaptation.”
> -> We have fixed this inconsistency in the text and now correctly refer only to offline training
>
> “the text in the main body does not mention that the reward model correlations for other tasks beyond half cheetah have are weaker with different patterns among the feedback types.”
> -> We have significantly expanded on the discussion of correlation and additional investigations of the reward function behavior
>
> “all feedback types individually can learn reward functions that closely match the ground-truth reward.”
> -> We have adapted this claim and now give more detailed conclusions and results
>
> “.no feedback type is more or less robust to noise.”
> -> Again, we have extended this discussion with updated results and figures. Importantly, we want to note that all feedback types behave relatively well with low noise levels and find that reward modeling performance (w.r.t. the reward function) does not necessarily translate to downstream RL performance.
>
> “Section 5 opens by stating that different feedback types struggle in different scenarios.”
> -> We sharpened this wording and acknowledge that it was vague in the previous version
>
> Formatting Issues
> -> We carefully went through the raised issues and tried to fix them for the revision

---

> > ### Comment · Reviewer_oTK8 · 2024-11-25
> > **Response to Authors Rebuttal**
> >
> > Thank you for your responses and the work you have done to update the paper. The extent to which the benchmark transfers to feedback from real humans is a big concern from me. In the absence of results validating that learnings can transfer, my worry is people will use this tool to reach conclusions and develop algorithms that have a massive sim-to-real gap. Therefore, I am not able to change my score.

---

> > > ### Author Response · Authors · 2024-11-28
> > >
> > > We appreciate the reviewer for sharing their honest concerns, as this has been very helpful in formulating the limitations and future work section of our manuscript.
> > >
> > > We would like to give some final points of consideration:
> > > - We try to openly acknowledge this limitation in our work, and the approach is well suited to integrate data from human annotations in the future, as well as enable experimentation in this area.
> > > - We have stated that the feedback dataset generation process is an important step which has an influence on downstream results. As a remedy, we have put considerable effort in the documentation of the feedback dataset (See Appendix B). We think that a transparent view of the dataset composition contributes to reproducible research. Existing work has sometimes struggled to provide these insights (e.g., distribution of prompts/queries/feedback values, etc.). We would like to contribute to setting a standard by clearly communicating the process, underlying assumptions, and resulting datasets, and provide tools for other researchers. We have tried to strengthen this contribution in the final revision by (1) Addressing this point more directly in the manuscript, and highlighting the tools for dataset analysis as core elements of the library, and (2) Extending the existing Appendix B, e.g. to document the effect of introduced noise.

---

> ### Comment · Reviewer_oTK8 · 2024-12-03
> **Response to Authors**
>
> Thank you for your response and the additional details included in the paper about the dataset generation process. However, the extra details do not address my concerns about the size of the sim-to-real gap between the conclusions people would draw using this toolkit and those that hold up in the “real world”.  For example, conclusions about the value and benefits of different sources of feedback (the main focus of Section 4). I read the section you added about real human data. My concern is not that real human data has to be included in the toolkit, but that the design decisions you have made lead to a small sim-to-real gap. There is great value in simulators, but the validation is important.

---

### Official Review · Reviewer_rXcY · 2024-11-03

**Soundness:** 3
**Presentation:** 3
**Contribution:** 3
**Rating:** 6
**Confidence:** 5

**Summary:**

The paper develops a lightweight software library for generating six feedback types—rating, comparative, demonstration, corrective, descriptive, and descriptive preference—in the field of Reinforcement Learning from Human Feedback (RLHF). The library is compatible with established RL frameworks, including Gymnasium, Stable Baselines, and Imitation.

The paper introduces noise into the synthetic feedback. Experimental results are presented on basic Gym Mujoco locomotion tasks. In these experiments, the authors compare the learned reward functions and the agent’s performance across different feedback types, both with and without noise. Additionally, the authors present a joint reward feedback method, i.e., an ensemble of rewards, and compare it to single feedback type baselines. This approach performs well in the HalfCheetah-v5 environment but is less effective in Walker2d-v5.

**Strengths:**

With the development of RLHF, there has been an exponential growth of research, especially in interdisciplinary applications, such as large language models (LLMs). I appreciate that it is important to have a standard library of feedback types commonly found in established RL frameworks. This is undoubtedly helpful for both new and experienced researchers in this rapidly evolving field. The choice of RL methods (e.g., PPO) and environments used in the paper are standard and widely accepted in the literature.

Furthermore, by encompassing multiple feedback types, this library has the potential to standardize RLHF research, making studies more comparable and reproducible across the field. Unlike prior work that often focuses on single feedback types or limited noise modeling, this toolkit provides a broader, more robust framework, which could foster progress in handling realistic feedback conditions in reinforcement learning.

**Weaknesses:**

1) Limited Analysis of Feedback Types and Noise Effects: The paper provides only a shallow analysis of the results across different feedback types and the impact of adding noise. The authors introduce Gaussian noise as a way to simulate realistic inconsistencies in human feedback, which is a valid approach. However, they assume that the added noise will uniformly challenge the agent’s learning process, yet they provide limited empirical support to demonstrate the nuanced effects of this noise. In reinforcement learning, robustness to noise is complex and context-dependent; simply adding noise doesn’t necessarily simulate real-world variability comprehensively. This is particularly relevant in cases where the agent encounters unfamiliar scenarios, as it may lack the generalization needed to adapt successfully, which is not addressed here. It would be beneficial if the authors quantified the noise across feedback types and analyzed how this type-specific noise impacts learning stability and performance. Without such quantification, comparing the robustness of different feedback types remains somewhat speculative and may not provide fair insights. As highlighted by Casper et al. (2023), inconsistencies in human feedback are a fundamental limitation in reinforcement learning from human feedback, underscoring the need for a systematic approach to evaluating robustness in such noisy environments.

Reference: Casper, Stephen, et al. "Open Problems and Fundamental Limitations of Reinforcement Learning from Human Feedback." Transactions on Machine Learning Research, 2023.

2) Reward Ensemble Approach is Underdeveloped: The paper attempts to improve reward learning through a reward ensemble (combining different feedback types), which is an intriguing direction. However, the ensemble approach lacks depth and does not yield substantial improvements. The authors themselves acknowledge that rewards from different feedback types cannot be averaged simply, and yet, the methods presented for combining them are fairly straightforward. Given the challenging nature of ensemble reward learning, the results are not strong enough to warrant this as a core contribution. As such, this component appears more exploratory than foundational, and it could benefit from either more sophisticated ensemble techniques or a more extensive evaluation to validate its potential utility.

3) Insufficient Details as a Software Library: As a contribution to reinforcement learning tools, the paper lacks critical details expected from a software library description. While the library supports diverse RL environments beyond the basic Gym Mojoco tasks, only these tasks are presented in the main text. Further, there is little mention of essential details such as hardware requirements, training time, or memory usage, which are crucial for reproducibility and practical use by researchers. Although some hyper-parameter details are provided in the appendix, this is insufficient for a full understanding of the library’s operational requirements and expected performance.

4) Typos and Formatting Issues

- Figure 4b title should read “RL Episode Returns.”

- Incomplete sentences in the paragraph near line 380.

- Hidden text in line 356.

**Questions:**

1) How diverse is the distribution of trajectories in line 197? Can you provide quantification of the exploration?

2) Is there any analysis for Figure 2, specifically why demonstrative and corrective feedback appear significantly different from other methods?

3) How can you ensure it is fair to compare different feedback types given that noise is generated separately for each type?

4) What are the hardware requirements and other support needed, along with the expected training time and latency based on these specifications?

5) Are there any results available for environments beyond Mujoco?

**Details Of Ethics Concerns:**

No concerns.

---

> ### Author Response · Authors · 2024-11-24
>
> We thank the reviewer for their comments and helpful suggestions. We have summarized the steps for a revised version in a summary comment. We still want to answer the points raised in this review briefly:
>
> “Limited Analysis of Feedback Types and Noise Effects”
> -> We have tried to improve the investigation of noise effects on the reward functions (Section 4.5) and now draw more robust conclusions: We find that reward functions show different drop-off behavior when increasing the noise level
>
> “Reward Ensemble Approach is Underdeveloped”
> -> We agree that this is not a fully developed method, and have made it more evident in our phrasing that the ensemble approach should be treated as a proof of concept, however, we have extended both details on the methods and provided additional results
>
> “Insufficient Details as a Software Library, What are the hardware requirements and other support needed, along with the expected training time and latency based on these specifications?”
> -> We have added some details on hardware requirements in the reproducibility statement and plan to add some additional code details for the final revision
>
> “Typos and Formatting Issues”
> -> We have tried to address these issues in the revision
>
> “How diverse is the distribution of trajectories in line 197? Can you provide quantification of the exploration?”
> -> We find that this method provides good coverage of the state space and will provide a figure in the appendix comparing it to random exploration
>
> “How can you ensure it is fair to compare different feedback types given that noise is generated separately for each type?”
> -> We tried to harmonize the noise generation scheme for each feedback type to ensure reproducibility (i.e., not using naive preference switching, which is not compatible with scalar reward functions), however, we acknowledge that the noise generation scheme for demonstrative feedback is indeed separate from the other feedback types
>
> “Are there any results available for environments beyond Mujoco?”
> -> We have provided additional results for Highway-Env and MetaWorld, which are broadly in line with existing experiments

---

> > ### Comment · Reviewer_rXcY · 2024-11-25
> >
> > Thank you for the detailed and thoughtful revisions. The additional experiments and clarifications comprehensively address my concerns, particularly regarding the analysis of noise effects, broader environments, and reproducibility details. These improvements significantly enhance the manuscript’s clarity, impact, and applicability, and I appreciate the authors’ efforts in strengthening the work. I have raised my score accordingly.

---

### Official Review · Reviewer_PQUp · 2024-11-04

**Soundness:** 3
**Presentation:** 2
**Contribution:** 3
**Rating:** 8
**Confidence:** 5

**Summary:**

This paper thoroughly investigates learning from different types of human feedback, including defining various types of human feedback, detailing how to collect synthetic feedback for research purposes, and training reward models based on synthetic feedback. It also analyzes the performance of joint training with multiple feedback types and examines the effectiveness of different feedback types.

**Strengths:**

1.Studying different types of human feedback is extremely important and can promote research development in the RLHF community.
2.Provides methods for generating various types of synthetic feedback for future RLHF research.
3.For the first time, it proposes training with multiple feedback sources while considering human noise.
4.The various analyses of reward models in the main paper and the appendix are comprehensive.

**Weaknesses:**

1.The workload of this paper is substantial, covering many key points, which results in relatively preliminary research on each type of feedback. The characteristics of different feedback types are not well demonstrated. Can you describe several key feedback types or explain which feedback types are more suitable for specific scenarios?
2.The first half of the paper is well-written, but the experimental organization in the latter half is chaotic, making it difficult to draw clear conclusions.

Overall, I believe this paper is highly valuable, but the current version appears somewhat hasty.

**Questions:**

1. Figure 3 obscures the text.
2. Some figures are not analyzed, and there are more important results in the appendix, requiring a restructuring of the paper.
3. Mean and standard deviation are not provided.
4. Why is the correlation of some reward functions so low?
5.Is the method for training reward models online? (i.e., continuously updated online with new samples)

---

> ### Author Response · Authors · 2024-11-24
>
> We thank the reviewer for their comments and helpful suggestions. We have summarized the steps for a revised version in a summary comment. We still want to answer the points raised in this review briefly:
>
> “Can you describe several key feedback types or explain which are more suitable for specific scenarios?”
> -> We have significantly expanded on the analysis of different feedback types in section 4, including a more detailed analysis of the relationship between reward model performance and environment and effectiveness for different scenarios.
>
> “The first half of the paper is well-written, but the experimental organization in the latter half is chaotic, making it difficult to draw clear conclusions.”
> -> We have reworked the second half to improve the readability and significance of the results
>
> “Figure 3 obscures the text.”
> -> We have tried to resolve all layouts and text issues
>
> “Some figures are not analyzed, and more important results are in the appendix, requiring a paper restructuring.”
> -> We have reworked the figures, in particular summarizing more key results without relying on the appendix for critical insights
>
> “Mean and standard deviation are not provided.”
> -> We have added these also in the main paper figures, although we want to note that they were consistently reported in the appendix and discarded for reasons of visual clarity before
>
> “Why is the correlation of some reward functions so low?”
> -> We have tried further investigating this in the paper (Figure 4 and Section 4.4). We conclude that low correlation is especially prevalent for corrective and demonstrative feedback, which rely on expert policies, i.e., are more independent from the ground-truth reward function; however, intriguingly, even very low-correlated reward functions can be very effective for training RL agents.
>
> “Is the method for training reward models online?”
> -> Our library has both capabilities; however, we have not reported any online training results in the paper, as it would add another layer of complexity. We have therefore removed references to online training to avoid confusion and now only refer to pre-training of reward functions

---

> ### Comment · Reviewer_PQUp · 2024-11-26
>
> I appreciate the authors' outstanding efforts during the rebuttal. It is gratifying to see that the quality of the revised manuscript has greatly improved, but there are still some unclear areas. I hope the authors can continue to check and optimize the manuscript in subsequent versions. Additionally, I have reviewed the opinions of other reviewers and believe that most of their concerns have also been addressed. I appreciate the authors' efforts on this work, and I have raised the score from 6 to 8 with confidence 5.
>
> Nonetheless, I still have a few minor questions: 1. Is the legend missing in Figure 2? I cannot understand this figure. 2. Considering this is a systematic paper, will all the code be open-sourced in the future? This will greatly enhance the impact and contribution of the work. 3. Have the authors attempted real human annotation? Annotating so much feedback could be costly. I suggest adding a more in-depth discussion on AI feedback and fine-grained feedback in future work, such as [1][2][3][4].
>
> [1] Liu J, Yuan Y, Hao J, et al. Enhancing Robotic Manipulation with AI Feedback from Multimodal Large Language Models[J]. arXiv preprint arXiv:2402.14245, 2024.
> [2] Wang Y, Sun Z, Zhang J, et al. Rl-vlm-f: Reinforcement learning from vision language foundation model feedback[J]. arXiv preprint arXiv:2402.03681, 2024.
> [3] Dong Z, Yuan Y, Hao J, et al. Aligndiff: Aligning diverse human preferences via behavior-customisable diffusion model[J]. arXiv preprint arXiv:2310.02054, 2023.
> [4] Lee H, Phatale S, Mansoor H, et al. Rlaif: Scaling reinforcement learning from human feedback with ai feedback[J]. 2023.

---

> > ### Author Response · Authors · 2024-11-26
> >
> > We thank the reviewer for their comments, and really appreciate the encouraging and productive discussion, which was crucial to improve our manuscript.
> >
> > We would like to answer your questions:
> > 1. Thank you for pointing that out, we will add a legend/schematic, and will improve the spacing between subfigures , to improve readability of the plot. This plot shows the final episode reward/success rates for all evaluated environments across the investigated feedback types and baseline methods (with the colored areas indicating min./max. performance of different seeds)
> > 2. Yes! We plan to contribute the code as a lightweight library for future research. As we outline in the future work section, we see multiple research opportunities and want to enable future research in this area by open-sourcing the code.
> > 3. Thank you for the suggestion! Enabling the integration of feedback from diverse human annotation sources is indeed one of our prime motivations. We see this work on reward model as one key component of AI systems for multi-type feedback (another being user interfaces). Multiple feedback types open up the possibility to learn from the most efficient, suitable and informative feedback. We therefore plan to ensure interoperability of our library to new and existing systems for the collection and processing of human feedback annotation data. We will gladly add a discussion of these point in our final revision.

---

> > > ### Comment · Reviewer_PQUp · 2024-11-27
> > >
> > > Thank you for the detailed response, good job!

---

### Meta-Review · Area_Chair_Tq4S · 2024-12-19

**Metareview:**

This paper presents a framework and toolkit for investigating learning from different types of human feedback in reinforcement learning settings, along with implementations for six feedback types and their corresponding reward models. The reviewers praised the paper's contributions in standardizing multi-feedback RLHF research and its thorough empirical analysis across environments. Through revisions, the authors have addressed major concerns by adding experiments on discrete action spaces, providing more detailed analysis of reward correlations with error bars, and improving documentation of dataset generation. While limitations remain regarding sim-to-real transfer and validation with actual human feedback, the paper makes a valuable contribution by establishing a foundation for studying multi-feedback RLHF and providing tools for future research.

 Given the paper's technical merit and the authors' thorough revisions, I recommend acceptance.

**Additional Comments On Reviewer Discussion:**

The major remaining limitation raised by Reviewer oTK8 regarding validation with real human feedback is a valid concern. However, I agree with other reviewers that this represents an important direction for future work rather than a critical flaw, given the significant technical challenges and resource costs involved in large-scale human feedback collection.
To maximize impact, I encourage the authors to **fulfill their commitment to open-source the codebase** with comprehensive documentation.
The current simulation-based framework makes meaningful contributions by establishing foundational tools and insights that can guide future research incorporating human feedback.

---

### Decision · Program_Chairs · 2025-01-22

Accept (Poster)